# Temporal Preference Optimization for Unsupervised Retrieval

**HyunJin Kim** [1]  **Jaejun Shim** [1]  **Young Jin Kim** [2] *  **JinYeong Bak** [1] *

## Abstract

Unsupervised dense retrievers offer scalability by learning semantic similarity from unlabeled documents via contrastive learning, but they struggle to capture the temporal relevance, retrieving semantically related but temporally misaligned documents–an important aspect when a document collection spans multiple time periods (*e.g.* retrieving documents from 2018-2025 for "Who is the president in 2019?" introduces temporal ambiguity.). Existing methods rely on supervised training with explicit timestamps, which are not always feasible. We propose TPOUR (*Temporal Preference Optimization for Unsupervised Retriever*), which uses our novel training method *Temporal Retrieval Preference Optimization* (TRPO). TRPO reinterprets preference learning in the temporal dimension, guiding the retriever to favor temporally aligned documents. TPOUR further generalizes to unseen time periods via interpolation in a learned time embedding, enabling continuous temporal alignment. Experiments on temporal information retrieval (T-IR), TPOUR outperforms both unsupervised and supervised baselines. Compared to Qwen-Embedding-8B, despite being about $72.7\times$ smaller, TPOUR Contriever improves average nDCG@5 by +4.04 (+12.15%) on explicit and +4.98 (+15.21%) on implicit queries. We provide our code at https://github.com/agwaBom/TPOUR.

## 1. Introduction

Document retrieval is the process of identifying relevant documents from document collections (Gao et al., 2024; Zhao et al., 2024a;b; Zhu et al., 2025; Li et al., 2025). It is widely used for various applications, including search engines (Brin & Page, 1998; Li et al., 2025), recommendation systems (Bobadilla et al., 2013; Zhang et al., 2019; Singh, 2023; Li et al., 2024), question answering (Karpukhin et al., 2020; Zhang et al., 2023a;b), and retrieval-augmented generation (Lewis et al., 2020; Zhao et al., 2024a; Fan et al., 2024; Kwon et al., 2025). Retrieval training generally falls into supervised and unsupervised methods. Supervised methods utilize labeled query-document pairs (Karpukhin et al., 2020), whereas unsupervised methods leverage term-frequency (Robertson & Zaragoza, 2009) or contrastive learning from unlabeled data (Izacard et al., 2022).

Despite advancements in retrieval research, most retrieval systems overlook *temporal misalignment* (*i.e.*, mismatch between the temporal context of user queries and the timestamps of retrieved documents). *Temporal retrieval* aims to address this limitation by incorporating temporal context into the retriever. As shown in Fig. 1, queries may contain *explicit* (*e.g.*, "in 2019") or *implicit* (*e.g.*, "this year") temporal information. While explicit references clearly anchor the query in time, implicit ones require interpretation. We adopt an approach that trains the retriever to prefer documents from a specific time period and interpret implicit queries accordingly. For example, a retriever trained on 2018 data would interpret "this year" as referring to 2018, aligning implicit temporal expressions with its training period. Temporal retrieval is important in domains such as news (Litty K Mathews, 2012; Wang et al., 2012; Luu et al., 2022) and law (Schilder & McCulloh, 2005), where the relevance of information depends on its publication date. For instance, the query "What was the minimum wage law in effect in 2019?" should retrieve the regulation in effect at that time.

However, existing retrieval methods often neglect temporal signals, particularly when timestamps are implicit rather than explicitly stated in the query. For instance, consider the query "Who is the current president?", which implicitly requires an answer at the time the query is raised, despite the absence of an explicit timestamp. Time-unaware retrievers such as Contriever (Izacard et al., 2022) or DPR (Karpukhin et al., 2020) are trained to maximize semantic similarity, and thus often retrieve documents that are semantically relevant but temporally unaligned. Fig. 1 illustrates this limitation–a time-unaware retriever fails to distinguish temporally aligned documents from temporally misaligned ones when relying solely on semantic similarity.

[1]Sungkyunkwan University, Suwon, South Korea [2]Microsoft, Redmond, USA. Correspondence to: Young Jin Kim <youki@microsoft.com>, JinYeong Bak <jy.bak@skku.edu>.

*Proceedings of the $43^{rd}$ International Conference on Machine Learning*, Seoul, South Korea. PMLR 306, 2026. Copyright 2026 by the author(s).

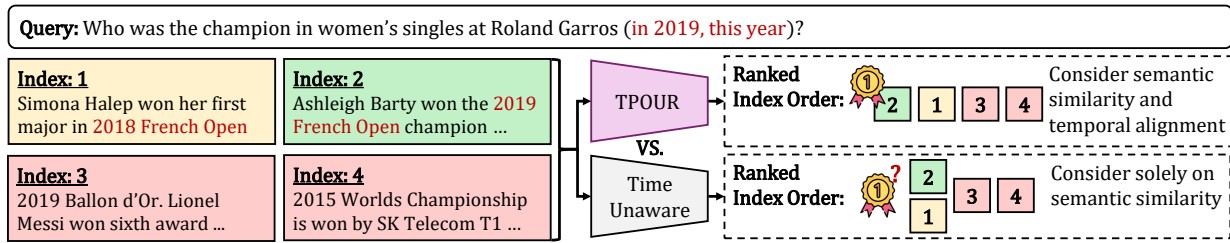

*Figure 1.* Comparison between TPOUR aligned at 2019 and a time-unaware retriever for queries with explicit (*e.g.*, in 2019) or implicit (*e.g.*, this year) temporal information. **Left**: A mixed-timestamp document collection containing (i) semantically and temporally aligned documents (green), (ii) semantically relevant but temporally misaligned documents (yellow), and (iii) irrelevant documents (red). **Right**: Ranked retrieval results. The time-unaware retriever, trained solely for semantic similarity, struggles to rank the temporally aligned document (green) over the misaligned (yellow). In contrast, the TPOUR-trained retriever prioritizes the temporally aligned document.

In practice, addressing temporal misalignment is challenging. On the one hand, supervised approaches may capture temporal relevance, but they require large amounts of labeled data, making them impractical at scale. On the other hand, unsupervised approaches based on contrastive learning (Shao et al., 2021; Izacard et al., 2022; Wu et al., 2022; Deng et al., 2022) are scalable but solely optimize for semantic similarity and ignore temporal relevance.

To embed temporal relevance in unsupervised retrieval, we propose TPOUR (*Temporal Preference Optimization for Unsupervised Retriever*), which integrates novel training method *Temporal Retrieval Preference Optimization* (TRPO) with contrastive learning. TRPO incorporates temporal preference signal into the retriever, reinterpreting preference learning in the temporal dimension using training signals from document corpora collected at different time periods. Rather than relying solely on semantic similarity, TRPO prioritize temporally aligned documents over misaligned ones. Thus, TPOUR preserves semantic similarity while learning temporal relevance, even when explicit time information is missing from the query or document.

TPOUR does not require retraining to adapt to specific time periods. We validate that the time vector, originally proposed as a temporal embedding for generative models (Nylund et al., 2024), can be applied to encoder-based TPOUR retriever. By extracting time vectors from TPOUR retrievers fine-tuned on a specific time period and interpolating them, we achieve continuous temporal alignment to intermediate periods without retraining. Our main findings are as follows:

1. **Temporal misalignment occurs in existing retrieval.** We show that time-unaware retrievers tend to retrieve semantically relevant but temporally misaligned documents from a document collection with mixed-timestamps.

2. **Integrating preference optimization helps capture temporal awareness.** We propose TPOUR, which learns to prefer temporally aligned over misaligned documents, improving temporal retrieval and enabling timestamp prediction using minimal corpus-level supervision.

3. **Time vectors enable continuous temporal generalization.** We validate that time-vector interpolation (Nylund et al., 2024) can be applied to TPOUR-trained retrievers, allowing them to generalize to intermediate time periods without additional training. We further show that extrapolation enables generalization to future time.

4. **Temporal awareness reveals time sensitivity in general retrieval tasks.** On the BEIR benchmark, TPOUR uncovers alignment between dataset publication year and optimal retrieval performance, suggesting that temporal modeling improves even general retrieval tasks.

## 2. Related Work

### 2.1. Unsupervised Learning for Retrieval Training

Unsupervised learning has enabled retrievers to scale with large amounts of unlabeled documents, from early statistical methods (Jatowt et al., 2005; 2013; Berberich et al., 2010; Kanhabua & Nørvåg, 2010; Kanhabua et al., 2012) like BM25 (Robertson & Zaragoza, 2009) to recent neural embedding models (Nussbaum & Duderstadt, 2025). While traditional approaches rely on statistics, unsupervised dense retrievers leverage contrastive learning. In dense retrieval, DPR (Karpukhin et al., 2020) is a supervised dense retriever trained on labeled query-passage pairs. In contrast, Contriever (Izacard et al., 2022) utilizes fully unsupervised contrastive learning. REALM (Guu et al., 2020) introduces retrieval-augmented masked language modeling. SimCSE (Gao et al., 2021) applies in-batch contrastive learning for sentence embeddings. E5 (Wang et al., 2024b) extends this with weak supervision over large-scale web data. CPT (Neelakantan et al., 2022) shows that scaling contrastive learning improves both text and code embeddings. GTE (Li et al., 2023) improves generalization by training on diverse datasets, while M3-Embedding (Chen et al., 2024) uses self-distillation to unify signals from multiple retrieval paradigms. Most recently, Nomic Embed v2 (Nussbaum & Duderstadt, 2025) adopts a sparse mixture-of-experts (MoE) for scalable and efficient general-purpose embedding.

Contrastive learning is the core of unsupervised retriever training, where a query $Q$ is paired with a positive document $D^+$ and a set of negative documents $\{D_1^-, ..., D_K^-\}$. The loss (Eq., 1) is calculated using a similarity function $S(\cdot, \cdot)$ with a query encoder $\pi_q$ and document (*i.e.*, key) encoder $\pi_k$. This loss encourages models to maximize similarity between a query and its positive document while minimizing similarity to negatives. However, embeddings are solely optimized for semantic similarity. As a result, retrievers such as Contriever (Izacard et al., 2022) degrade in mixed-timestamp document collection settings, failing to distinguish between documents from different time periods. Let $q = \pi_q(Q)$ and $d = \pi_k(D)$ denote the query and document embeddings, respectively. The contrastive loss is:

$$\mathcal{L}_{\text{CE}} = -\log \frac{\exp\big(S(q, d^+)\big)}{\exp\big(S(q, d^+)\big) + \sum_{i=1}^{K} \exp\big(S(q, d_i^-)\big)} \quad (1)$$

Unsupervised retrieval training commonly utilizes either (1) in-batch negative (Lee et al., 2019), or (2) MoCo (Momentum Contrast) (He et al., 2020). The former is effective with large batch sizes, while MoCo simulates large batches with lower memory. In MoCo, the query encoder $\pi_q$ and key encoder $\pi_k$ are updated during training. After updating $\pi_q$'s weight $\theta_q$ via the contrastive loss in Eq. 1, the key encoder weight $\theta_k$ is updated via momentum $\theta_k \leftarrow m \times \theta_k + (1-m) \times \theta_q$. In this work, we adopt MoCo for unsupervised retrieval to train under limited resources.

### 2.2. Temporal Relevance Modeling

Temporal relevance has been explored in language models (Lazaridou et al., 2021; Röttger & Pierrehumbert, 2021; Rosin et al., 2022; Su et al., 2023; Wang et al., 2023). For instance, (Dhingra et al., 2022) jointly models timestamps with text to improve temporal generalization in language modeling. In temporal information retrieval, recent work incorporates temporal information for time-aware search (Wu et al., 2024; Abdallah et al., 2025). For example, (Gade et al., 2025) applies retrieval-augmented generation on explicit temporal annotation for both queries and documents, and (Qian et al., 2024) addresses implicit temporal awareness through query rewriting over a knowledge graph.

Another line of work extracts time vectors from generative language models fine-tuned on data from distinct periods (Nylund et al., 2024). These latent vectors capture temporal context and allow interpolation. They show that adjacent time vectors are close in weight space, enabling generalization to intermediate periods without retraining. We extend time vector extraction from generative language models to TPOUR, enabling continuous temporal alignment of retrievers to unseen intermediate and future periods.

### 2.3. Direct Preference Optimization

RLHF (Reinforcement Learning from Human Feedback) aligns language models with human preferences (Ouyang et al., 2022). It involves training a reward model on human-labeled preferences and optimizing the policy $\pi_\theta$ to maximize the reward using PPO (Proximal Policy Optimization) (Schulman et al., 2017) or DPO (Direct Preference Optimization) (Rafailov et al., 2023).

$$\mathcal{L}_{\text{DPO}} = -\log \sigma\Big(\beta \log \frac{\pi_\theta(y^w \mid x)\, \pi_{\text{ref}}(y^l \mid x)}{\pi_\theta(y^l \mid x)\, \pi_{\text{ref}}(y^w \mid x)}\Big) \quad (2)$$

Building on DPO, we introduce TRPO, which incorporates temporal preferences into unsupervised retrieval. TRPO constructs preference pairs from document corpora across time and learns to prefer temporally aligned documents without explicit supervision. Unlike DPO, which aligns generation policies using human preferences, TRPO adapts preference optimization to retrieval by replacing log-likelihoods with embedding similarity from unlabeled temporal signals.

## 3. Temporal Preference Optimization for Unsupervised Retriever

### 3.1. Incorporating Temporal Preferences into Contrastive Learning

We propose TPOUR (Temporal Preference Optimization for Unsupervised Retriever), a training framework that integrates temporal preferences into contrastive learning for unsupervised retrieval. Built upon MoCo, TPOUR jointly learns semantic similarity and temporal relevance by combining contrastive learning with a preference-based objective from TRPO. This enables the retriever to encode both content relevance and implicit temporal preferences from unlabeled data. We illustrate this with a case study on unlabeled document training in Appendix E.

As shown in Fig. 2, the training phase consists of a query document $Q_i$, a temporally aligned document $D_i^t$, and an unaligned document $D_i^{t'}$. The encoder $\pi_\theta$ encodes these inputs, while a momentum-based reference encoder $\pi_{\text{ref}}$ maintains a queue of negatives for contrastive learning. The training objective combines two losses. The first is a contrastive loss that brings the query closer to its relevant document while distinguishing it from negatives, where $S(\cdot, \cdot)$ is the similarity function. Here, we define $S_\theta(y_i^w) = S(\pi_\theta(Q_i), \pi_\theta(D_i^t))$, which denotes similarity with the temporally aligned document (preferred) and $S_\theta(y_i^l) = S(\pi_\theta(Q_i), \pi_\theta(D_i^{t'}))$, the similarity with the unaligned document (less preferred). The values $S_{\text{ref}}(y_j^w)$ and $S_{\text{ref}}(y_j^l)$ correspond to negative pairs from the previous batch queue $j$, where $D_j^- \in \{D_j^t, D_j^{t'}\}$:

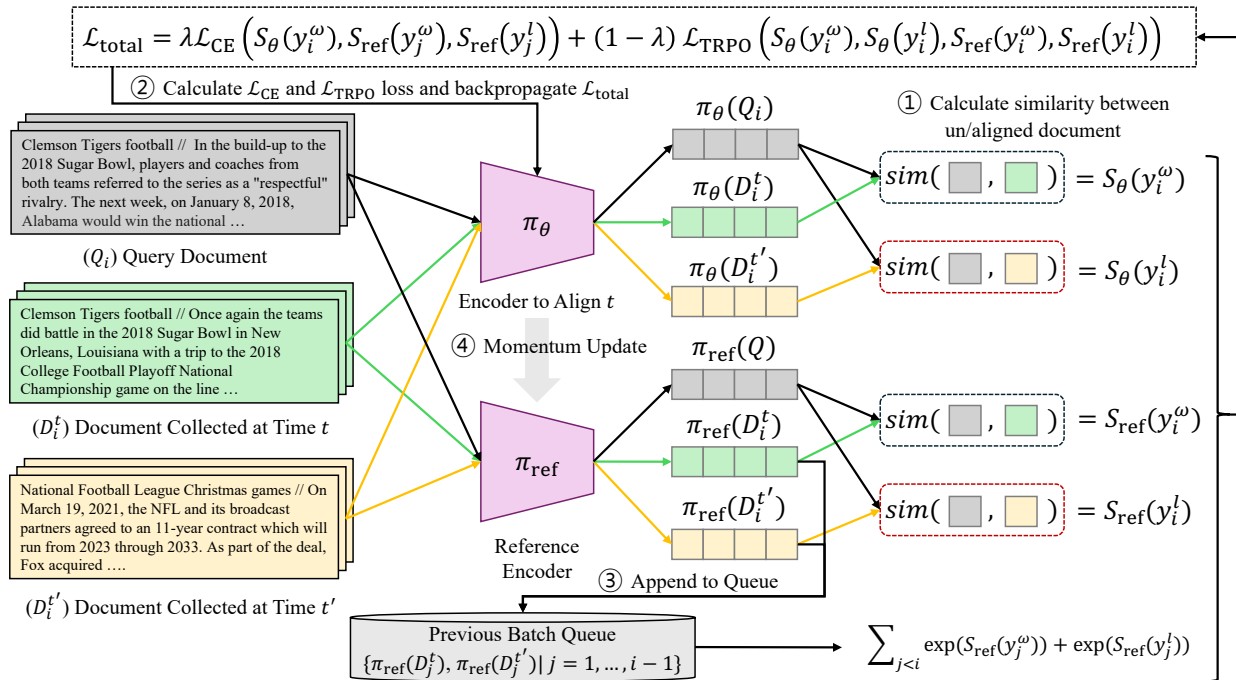

$$\mathcal{L}_{\text{total}} = \lambda \mathcal{L}_{\text{CE}} \left( S_\theta(y_i^\omega), S_{\text{ref}}(y_j^\omega), S_{\text{ref}}(y_j^l) \right) + (1 - \lambda) \, \mathcal{L}_{\text{TRPO}} \left( S_\theta(y_i^\omega), S_\theta(y_i^l), S_{\text{ref}}(y_i^\omega), S_{\text{ref}}(y_i^l) \right)$$

*Figure 2.* Overview of TPOUR. Given a query $Q_i$ and two documents $D_i^t$ (temporally aligned) and $D_i^{t'}$ (temporally misaligned), each input is encoded using both the main encoder $\pi_\theta$ and the reference encoder $\pi_{\text{ref}}$. ① Similarity scores are computed between the query and each document using $\pi_\theta$. ② A contrastive loss $\mathcal{L}_{\text{CE}}$, which calculate semantic similarity between $Q_i$ and $D_i^t$, and a TRPO loss $\mathcal{L}_{\text{TPRO}}$ for preferring temporally aligned documents are calculated to get combined loss $\mathcal{L}_{\text{total}}$. ③ The reference embeddings $\pi_{\text{ref}}(D_i^t)$ and $\pi_{\text{ref}}(D_i^{t'})$ are added to a queue as negatives for future batches. ④ The encoder $\pi_\theta$ is updated via $\mathcal{L}_{\text{total}}$, and $\pi_{\text{ref}}$ is updated via momentum from $\pi_\theta$.

$$\mathcal{L}_{\text{CE}} = -\log \frac{e^{S_\theta(y_i^w)}}{e^{S_\theta(y_i^w)} + \sum_{j<i} \left\{ e^{S_{\text{ref}}(y_j^w)} + e^{S_{\text{ref}}(y_j^l)} \right\}} \quad (3)$$

To model temporal preferences, TRPO aligns the preference gap between the current and reference models, where $S_\theta(y)$ and $S_{\text{ref}}(y)$ denote scores from the current and reference models given output $y$. Given a pair $y_i^w$ (preferred) and $y_i^l$ (less preferred), the TRPO loss is defined as Eq. 4. A detailed theoretical basis of TRPO is in Appendix B.1.

$$\begin{aligned} \mathcal{L}_{\text{TRPO}} = -\log \sigma \Big( \beta \big[ &S_\theta(y_i^w) - S_\theta(y_i^l) \\ &- \big( S_{\text{ref}}(y_i^w) - S_{\text{ref}}(y_i^l) \big) \big] \Big) \end{aligned} \quad (4)$$

The total loss is computed as $\mathcal{L}_{\text{total}} = \lambda \mathcal{L}_{\text{CE}} + (1-\lambda) \mathcal{L}_{\text{TRPO}}$, where $\lambda \in [0, 1]$ balances the influence of semantic and temporal signals. The encoder $\pi_\theta$ is optimized using $\mathcal{L}_{\text{total}}$, while the reference encoder weights $\theta_{\text{ref}}$ are updated via momentum as $\theta_{\text{ref}} \leftarrow m \times \theta_{\text{ref}} + (1 - m) \times \theta$, where $m$ is the momentum coefficient and $\theta$ is the current weight of $\pi_\theta$. After training, TPOUR-trained retrievers can be used as general-purpose retrieval systems through the standard inference pipeline, as illustrated in Appendix Fig. 6.

## 3.2. Continuous Temporal Representation

Discrete temporal models are inherently limited in modeling continuous time. Since time is inherently continuous, a retriever needs to generalize to queries that fall between the temporal regions covered by separately trained retrievers. To overcome this limitation, we adopt time vector extraction from language modeling (Nylund et al., 2024) and extend it to TPOUR for unsupervised retrieval.

We extract time vectors from TPOUR-trained retrievers fine-tuned on specific time periods (*e.g.*, the years 2018 and 2021). Interpolating between these vectors allows the model to adjust its temporal alignment and generalize to intermediate periods without retraining. Tab. 3, Fig. 3, and Fig. 4 show the generalization capability through time vector interpolation across continuous time shifts.

Formally, let $\theta_{\text{base}}$ denote the base encoder weight and $\theta_t$ the encoder weight fine-tuned on data from time period $t$. The time vector $\tau_t$ for time period $t$ is computed as $\tau_t = \theta_t - \theta_{\text{base}}$, where $\tau_t$ captures the temporal shift between the base model and the model adapted to time period $t$. To obtain an encoder for an intermediate time period $t_{\text{mid}}$, given two time vectors $\tau_{t_{\text{start}}}$ and $\tau_{t_{\text{end}}}$ corresponding to the $t_{\text{start}}$ (earlier) and $t_{\text{end}}$ (later), respectively, we interpolate using a coefficient $\alpha \in [0, 1]$, as defined in Eq. 5. Further

theoretical details are provided in Appendix Sec. B.2.

$$\theta^{t_{\text{mid}}} = \theta_{\text{base}} + (1 - \alpha)\tau_{t_{\text{start}}} + \alpha\tau_{t_{\text{end}}},$$
$$\text{where } t_{\text{start}} \leq t_{\text{mid}} \leq t_{\text{end}} \tag{5}$$

This interpolation allows the model to adjust its temporal alignment without retraining. For example, interpolating between 2018 and 2021 vectors enables retrieval for queries from 2019 or 2020. Tab. 12, Tab. 13, and Tab. 14 in the Appendix show that interpolation improves generalization to intermediate time periods even when the temporal information is not given in the query.

### 3.3. Inferring Document Timestamps from `TPOUR`

In addition to the retrieval, `TPOUR` can be used to infer a document's timestamp. Following (Gunasekaran et al., 2023), we formulate timestamp inference as a classification task and introduce a timestamp predictor based on a mixture of `TPOUR` retrievers (mixture-of-`TPOUR`). As illustrated in Appendix Fig. 7, the mixture-of-`TPOUR` uses a set of frozen retrievers $\pi_\theta^{t_1}, \ldots, \pi_\theta^{t_n}$, each specialized for a distinct time period $t_i$. Given a document $D$, each retriever encodes $D$ into a temporally-aware embedding. These embeddings are concatenated and passed to a shared trainable linear classification head to train and predict the timestamp.

We compare against a baseline predictor that uses a single frozen retriever $\pi_\theta$ trained on the full time range. To ensure a fair comparison, we match the number of trainable parameters by stacking multiple linear layers in the baseline classifier, with the same depth as the number of retrievers in the mixture model, and also compare model with larger parameter counts. The result shows mixture-of-`TPOUR` achieves superior timestamp prediction performance (Tab. 4).

## 4. Experiments and Analysis

This section presents experiments to answer three main research questions regarding `TPOUR`:

**RQ1. Do `TPOUR`-trained retrievers learn temporally aligned representations?** We evaluate whether `TPOUR`-trained retrievers retrieve temporally aligned documents and whether interpolation and timestamp prediction reveal embedded temporal representations in the retriever.

**RQ2. Does temporal awareness improve performance on temporal QA tasks?** We assess temporal awareness by evaluating retrieval on temporal QA across time splits and measuring gains in periods via time vector interpolation.

**RQ3. Can temporal awareness reveal time sensitivity in general retrieval tasks?** We conduct a case study on the BEIR benchmark (Thakur et al., 2021), spanning diverse domains and publication years, to assess whether temporal awareness in `TPOUR`-trained models reveals time sensitivity.

*Table 1.* Evaluation bias test on SituatedQA. To confirm that the dataset construction process is free from bias in benchmark creation, we built a separate gold document collection with Nomic Embed v2 MoE and DPR as the retriever and averaged their results. The performance trends of `TPOUR` Contriever (2018/2021) remain consistent, showing that retriever does not affect benchmark bias.

| SituatedQA | 2018/N@5 | 2018/N@10 | 2021/N@5 | 2021/N@10 |
|---|---|---|---|---|
| Contriever | 31.28 | 34.94 | 35.21 | 38.13 |
| DPR (Dense Passage Retriever) | 27.21 | 31.13 | 33.82 | 36.37 |
| Nomic Embed v2 MoE | 31.30 | 34.74 | 34.52 | 36.96 |
| `TPOUR` Contriever (2018) | **38.70** | **41.13** | 11.57 | 14.17 |
| `TPOUR` Contriever (2021) | 23.15 | 27.63 | **42.93** | **43.62** |

### 4.1. Evaluation Benchmarks and Metrics

To assess `TPOUR`, we use two temporal QA datasets, SituatedQA and RealTimeQA, for temporal retrieval, and the BEIR for general retrieval tasks. (1) **SituatedQA** (Zhang & Choi, 2021) is a yearly temporal QA dataset containing 2,795 queries spanning 1700–2021. Since years prior to 2018 each have fewer than 130 queries, we focus on the 2018–2021 subsets, which contain 291, 411, 501, and 491 queries, respectively. (2) **RealTimeQA** (Kasai et al., 2023) is a monthly temporal QA dataset, providing weekly evaluations from June 2022 to January 2024, with approximately 130 queries per month. For evaluation, we use the queries from January to December 2023. (3) **BEIR** (Thakur et al., 2021) is a general retrieval benchmark comprising 18 datasets across diverse domains (*e.g.*, medical, financial). We use BEIR to show that temporal awareness reveals time sensitivity in general retrieval tasks.

SituatedQA provides only queries and associated answers, while RealTimeQA includes a query, a single associated document, and an answer, which is still insufficient for evaluating retrieval performance, since duplicated documents created or updated at different timestamps are not present. To address this, we construct a custom retrieval benchmark based on these datasets, following the BEIR custom dataset guidelines (Thakur, 2022) to create a temporal QA benchmark tailored for retrieval evaluation. Each custom dataset requires a set of documents related to each query. To construct these, we use Contriever (Izacard et al., 2022) to retrieve the top-10 documents per query from a fixed document collection. For instance, when building the document set for queries from the 2018 test set, we use the 2018 Wikipedia document collection, retrieve the top-10 documents using Contriever (Izacard et al., 2022), filter out documents that do not contain the answer, retaining only the answer-containing ones as gold documents. We also perform an evaluation bias test with a different retriever (DPR, Nomic-Embed v2 MoE) to check whether the performance trends remain, as reported in Tab. 1.

We evaluate retrieval performance using normalized discounted cumulative gain (nDCG@$k$, denoted as N@$k$), which captures relevance and ranking in the top-$k$.

Recall@$k$, the percentage of queries with at least one correct document in the top-$k$, is reported in Appendix D. For timestamp prediction, we report accuracy, which is the ratio of correct predictions to total examples.

## 4.2. Training Datasets

We construct our training corpus from English Wikipedia database dumps (Johnson et al., 2024) collected at different times to capture temporal differences, retaining newly added or modified document content across the corpus. For the yearly corpus, we use Wikipedia dumps from December 2018 and 2021, which serve as the yearly time span used for SituatedQA. For the monthly corpus, we use dumps from January and December 2023 for RealTimeQA. An additional dump is also used for temporal diversity. To prevent data leakage, we filter out documents that serve as gold documents in SituatedQA and RealTimeQA. Details on training data construction are provided in Appendix A.2 and Tab. 7, and the training setup in Appendix A.3.1.

## 4.3. Baselines

We consider three types of baselines. **Standard Retrievers** are retrieval models that rank documents primarily based on semantic relevance between the query and document. **Temporal-Aware Retrievers** are retrieval models that incorporate temporal signals, such as timestamps or temporal constraints, to better align retrieved documents with the time specified or implied by the query. **Large Embedding Models** are recent large-scale embedding models trained with broad instruction-following and retrieval objectives, which provide strong general-purpose retrieval performance and can potentially handle temporal intent through their pretrained representations or prompting. Detailed information about each baseline is provided in Appendix C.2.

**Standard Retriever:** (1) **DPR** (Karpukhin et al., 2020) is a supervised bi-encoder with 110M parameters, trained with BM25 hard negatives. (2) **REALM** (Guu et al., 2020) is a 134M parameter retriever that combines retrieval with language modeling in an end-to-end setup. (3) **SimCSE** (Gao et al., 2021) is a 110M parameter retriever that learns sentence embeddings via contrastive learning and can be adapted for retrieval. (4) **Contriever** (Izacard et al., 2022) is an unsupervised retriever with 110M parameters, trained via MoCo-based contrastive learning.

**Time-Aware Retriever:** (1) **Berberich et al. (2010)** is an early probabilistic model that explored temporal expressions represented as tuples. (2) **Temporal Contrastive** is a temporal-aware contrastive baseline that augments the standard contrastive retrieval objective with temporal supervision. We include this baseline to examine whether temporal alignment can be obtained by directly learning time-based positive and negative documents, without preference-based optimization (*i.e.* TRPO). For each query, temporally aligned documents are treated as positives and misaligned documents as negatives, yielding $\mathcal{L}_{\text{TempCE}}$, which encourages higher similarity between the query and documents that better match the target time. The final objective is $\mathcal{L}_{\text{total}} = \lambda \mathcal{L}_{\text{CE}} + (1 - \lambda)\mathcal{L}_{\text{TempCE}}$, where $\mathcal{L}_{\text{CE}}$ models semantic relevance and $\mathcal{L}_{\text{TempCE}}$ models temporal alignment. (3) **TimeR**[4] (Qian et al., 2024) proposes a time-aware retriever with 113M parameters, trained on temporal knowledge graphs. We use their public checkpoint for comparison.

**Large Embedding Model:** (1) **Nomic Embed v2 MoE** (Nussbaum & Duderstadt, 2025) is a recent general-purpose embedding model with 475M parameters, utilizing a sparse mixture-of-experts architecture. (2) **Qwen-3-Embedding-8B** (Zhang et al., 2025) is a large-scale embedding model with 8B parameters, built upon Qwen3 (Yang et al., 2025). It supports diverse embedding and reranking tasks across multiple domains and languages. We consider retrieval with naive, query rewriting (QR), and time-aware instruction retrieval (TAI), since Qwen-3-Embedding supports instruction-conditioned embeddings. We apply TAI to explicit temporal information, as its target time can be directly incorporated in the instruction. We report our prompt used for TAI in the Appendix Tab. 10.

## 4.4. Results and Analysis

### 4.4.1. DO TPOUR-TRAINED RETRIEVERS LEARN TEMPORALLY ALIGNED REPRESENTATIONS?

We evaluate whether TPOUR learns temporally aligned representations by analyzing the document timestamps distribution and timestamp prediction. Fig. 3 shows that interpolation $\alpha$ smoothly shifts retrieval distributions toward intermediate time periods. Full distributions for SituatedQA and RealTimeQA are in Appendix Fig. 8 and 9. Notably, TPOUR also captures temporal patterns without explicit supervision (Appendix Sec. D.7; Tab. 18).

To further assess whether TPOUR encodes temporal information, we evaluate its timestamp prediction accuracy as a classification task, with year prediction as 4 classes (2018–2021) and month prediction as 12 classes. As shown in Tab. 4, the mixture-of-TPOUR achieves 76.56% year accuracy and 27.41% month accuracy, outperforming the baseline predictor built on Contriever (50.18% year, 22.22% month accuracy) with 10,000 training steps. The evaluation loss also decreases from 3.13 to 2.66. Furthermore, Mixture-of-TPOUR with 2 encoders (220M) surpasses the larger-capacity Nomic-Embed v2 MoE (305M), achieving a 21.53% improvement on year accuracy and 20.30% on month accuracy. These results indicate that TPOUR embeddings preserve temporal signals for inference tasks that are both temporally aligned and predictive. The detailed evaluation setup is in Appendix Fig. 7.

*Table 2.* Retrieval performance on mixed-timestamp document collections across SituatedQA and RealTimeQA. We compare standard baselines using their public checkpoints against the TPOUR-trained retriever with three training seeds (mean and standard deviation reported with ±), denoted as TPOUR Contriever ($t$). TPOUR Contriever outperforms the baselines across time periods, achieving higher accuracy and stronger generalization regardless of whether queries contain explicit or implicit temporal information. Notably, TPOUR Contriever achieves strong performance on intermediate periods (2019, 2020, and June) without requiring time-specific retraining.

| Retriever | SituatedQA | | | | | | | | RealtimeQA | | | | | |
| | 2018 | | 2019 | | 2020 | | 2021 | | January | | June | | December | |
| | N@5 | N@10 | N@5 | N@10 | N@5 | N@10 | N@5 | N@10 | N@5 | N@10 | N@5 | N@10 | N@5 | N@10 |
|---|---|---|---|---|---|---|---|---|---|---|---|---|---|---|
| *Query with Explicit Temporal Information* | | | | | | | | | | | | | | |
| Contriever | 29.30 | 33.35 | 29.67 | 34.49 | 31.25 | 35.77 | 37.85 | 41.05 | 21.76 | 22.36 | 33.04 | **33.12** | 45.96 | 43.99 |
| REALM | 22.37 | 21.57 | 12.26 | 14.04 | 13.92 | 15.23 | 9.34 | 10.03 | 14.66 | 14.22 | 18.23 | 16.83 | 19.57 | 17.83 |
| SimCSE | 25.17 | 27.56 | 19.62 | 22.57 | 17.84 | 19.71 | 18.25 | 20.94 | 18.40 | 17.98 | 21.46 | 21.73 | 19.18 | 19.56 |
| Supervised:DPR | 28.67 | 31.20 | 27.58 | 30.76 | 27.91 | 31.24 | 32.76 | 34.62 | 22.90 | 22.55 | 30.27 | 29.03 | 35.65 | 34.49 |
| Berberich et al. (2010) | 8.64 | 9.41 | 9.36 | 10.15 | 8.48 | 9.63 | 9.91 | 10.88 | 15.84 | 16.33 | 8.61 | 8.74 | 22.47 | 24.91 |
| Temporal Contrastive | 35.00 | 38.60 | 29.03 | 34.76 | 29.21 | 33.72 | 35.97 | 37.39 | 25.69 | 25.19 | **33.99** | 32.22 | 39.88 | 37.12 |
| TimeR[4] | 33.65 | 37.71 | 27.62 | 32.24 | 31.09 | 34.71 | 31.33 | 34.97 | 26.45 | 25.34 | 31.36 | 29.47 | 8.94 | 8.60 |
| Nomic Embed v2 MoE | 29.61 | 33.62 | 29.67 | 34.71 | 30.77 | 35.17 | 31.09 | 33.74 | 22.38 | 22.12 | 31.45 | 30.64 | 36.99 | 36.28 |
| Qwen3-Embedding-8B | 30.45 | 33.77 | 32.77 | 35.26 | 36.31 | 40.06 | 35.17 | 37.85 | 22.54 | 23.30 | 33.86 | 33.00 | 41.06 | 39.07 |
| TPOUR Contriever (2018) | **43.93**±0.3 | **46.66**±0.1 | 31.25±0.4 | 34.11±1.1 | 24.56±2.5 | 27.44±2.7 | 18.25±3.8 | 21.58±3.5 | — | — | — | — | — | — |
| TPOUR Contriever (2021) | 24.38±3.0 | 29.06±3.1 | 23.63±3.4 | 27.59±3.8 | 26.60±1.7 | 29.68±1.9 | **40.21**±0.7 | **44.72**±6.8 | — | — | — | — | — | — |
| TPOUR Contriever (Jan) | — | — | — | — | — | — | — | — | **31.78**±0.5 | **31.37**±0.7 | 31.96±2.2 | 31.05±2.1 | 31.01±1.5 | 30.51±1.5 |
| TPOUR Contriever (Dec) | — | — | — | — | — | — | — | — | 9.21±0.7 | 9.51±0.3 | 27.43±1.4 | 26.30±1.4 | **48.24**±1.6 | **45.57**±1.0 |
| TPOUR Contriever | **43.93**±0.3 | **46.66**±0.3 | **32.87**±0.1 | **36.75**±0.3 | 30.56±1.1 | 33.66±0.9 | **40.21**±0.7 | **44.72**±6.8 | **31.78**±0.5 | **31.37**±0.7 | 32.82±0.4 | 31.90±0.4 | **48.24**±1.6 | **45.57**±1.0 |
| *Query with Implicit Temporal Information* | | | | | | | | | | | | | | |
| Contriever | 29.89 | 34.60 | 30.96 | 36.20 | 31.00 | 34.43 | 33.06 | 37.08 | 27.38 | 28.48 | 32.46 | 32.83 | 40.03 | 38.77 |
| REALM | 22.41 | 22.09 | 13.22 | 14.74 | 15.35 | 16.29 | 10.41 | 11.09 | 16.44 | 16.12 | 19.86 | 18.39 | 19.95 | 18.30 |
| SimCSE | 24.31 | 27.26 | 22.66 | 26.42 | 22.67 | 24.36 | 20.73 | 23.68 | 23.00 | 23.08 | 26.37 | 26.75 | 22.95 | 23.83 |
| Supervised:DPR | 32.75 | 35.18 | 30.31 | 34.45 | 28.46 | 31.71 | 31.17 | 34.29 | 29.21 | 28.38 | 27.87 | 28.38 | 33.82 | 32.45 |
| Berberich et al. (2010) | 9.14 | 9.35 | 8.14 | 9.37 | 7.43 | 8.19 | 8.05 | 8.68 | 13.21 | 13.35 | 7.13 | 7.79 | 20.43 | 22.91 |
| Temporal Contrastive | 34.64 | 37.18 | 33.45 | 36.81 | 31.13 | 35.12 | 33.59 | 37.78 | 31.58 | 30.26 | **34.62** | **33.44** | 39.55 | 37.76 |
| TimeR[4] | 35.50 | 39.52 | 28.43 | 32.90 | 32.02 | 36.17 | 30.86 | 34.22 | 26.95 | 26.08 | 8.79 | 8.38 | 32.76 | 32.52 |
| Nomic Embed v2 MoE | 29.23 | 33.27 | 30.61 | 33.81 | 30.51 | 34.86 | 30.42 | 33.08 | 25.78 | 25.88 | 30.66 | 31.37 | 35.82 | 34.94 |
| Qwen3-Embedding-8B | 28.07 | 32.60 | 33.27 | 33.52 | 35.91 | 39.57 | 33.91 | 36.45 | 24.25 | 24.40 | 32.25 | 32.57 | 41.51 | 38.30 |
| TPOUR Contriever (2018) | **44.11**±0.5 | **46.59**±0.2 | 31.23±0.4 | 34.49±0.5 | 24.80±2.1 | 27.87±2.0 | 18.53±3.3 | 21.87±2.9 | — | — | — | — | — | — |
| TPOUR Contriever (2021) | 24.81±2.6 | 29.47±2.8 | 26.48±1.6 | 30.52±1.3 | 28.31±1.3 | 31.84±1.8 | **39.40**±1.0 | **44.72**±6.8 | — | — | — | — | — | — |
| TPOUR Contriever (Jan) | — | — | — | — | — | — | — | — | **32.03**±0.8 | **31.67**±1.2 | 31.97±2.2 | 31.37±1.5 | 30.48±2.4 | 30.25±2.0 |
| TPOUR Contriever (Dec) | — | — | — | — | — | — | — | — | 10.34±1.3 | 10.89±2.2 | 28.73±4.2 | 27.58±3.6 | **49.29**±3.4 | **46.19**±2.1 |
| TPOUR Contriever | **44.11**±0.5 | **46.59**±0.2 | 34.36±0.2 | 38.63±0.4 | 33.14±0.2 | 36.31±0.5 | **39.40**±1.0 | **44.72**±6.8 | **32.03**±0.8 | **31.67**±1.2 | 31.73±0.9 | 32.86±0.4 | **49.29**±3.4 | **46.19**±2.1 |

*Table 3.* Interpolation of TPOUR Contriever between $t_{start}$ and $t_{end}$ periods reduces temporal misalignment in intermediate periods. The result shows (1) interpolation enables generalization in middle time ($\alpha = 0.5$). And (2) it can surpass directly fine-tuned retriever (Eval-year fine-tuned vs. Best interpolation $\alpha$).

| Method | SituatedQA | | RealTimeQA | |
| | nDCG@5 | nDCG@10 | nDCG@5 | nDCG@10 |
|---|---|---|---|---|
| TPOUR Contriever ($t_{start}$) | 29.05 | 32.13 | 30.29 | 29.87 |
| TPOUR Contriever ($t_{end}$) | 31.72 | 34.30 | 29.32 | 27.97 |
| $\alpha = 0.5$ | 35.71 | 39.23 | 30.47 | 30.36 |
| Best Interpolation $\alpha$ | **42.47** | **44.59** | **38.77** | **37.30** |
| Eval-year fine-tuned | **42.47** | 43.96 | 37.30 | 36.98 |

*Table 4.* Performance of the mixture-of-TPOUR timestamp predictor after 10k training steps. The mixture-of-TPOUR model achieves the lowest evaluation loss (Eval Loss), as well as the highest year accuracy (Y-Acc) and month accuracy (M-Acc). It also outperforms the larger size Nomic-Embed v2 MoE (305M) when compared to a mixture-of-TPOUR with two encoders (220M).

| | Eval Loss ↓ | M-Acc ↑ | Y-Acc ↑ |
|---|---|---|---|
| Contriever | 3.13 | 22.22 | 50.18 |
| Nomic-Embed v2 MoE | 3.54 | 5.57 | 53.03 |
| Mixture-of-TPOUR (2 Encoders) | 2.76 | 25.87 | 74.56 |
| Mixture-of-TPOUR (10 Encoders) | **2.66** | **27.41** | **76.56** |

### 4.4.2. DOES TEMPORAL AWARENESS IMPROVE PERFORMANCE ON TEMPORAL QA TASKS?

We evaluate the impact of temporal awareness on retrieval using SituatedQA and RealTimeQA. Tab. 2 shows nDCG@5/10 across different test periods. For interpolated TPOUR Contriever (denoted as TPOUR Contriever), we apply a heuristic interpolation strategy, selecting the interpolated model whose $\alpha$ corresponds to the $t_{mid}$ of the test set. For example, for the 2019 set, we use the interpolated model with $\alpha = 0.3$. The TPOUR Contriever consistently outperforms all baselines. On SituatedQA 2018 (Implicit), TPOUR achieves an nDCG@5 of 44.11, substantially surpassing Contriever (29.89). Similar improvements are observed across later years, including +3.40 nDCG@5 in 2019 and

+6.34 in 2021 over Contriever. On RealTimeQA, TPOUR also maintains an advantage across months in January and December. Notably, performance gains remain consistent across time periods, regardless of whether temporal information is provided explicitly or implicitly in the query.

Tab. 5 further compares TPOUR Contriever with Qwen3-Embedding-8B, a substantially larger embedding model, under query rewriting (QR) and time-aware instruction (TAI) settings. Although QR improves Qwen3-Embedding-8B on the 2018 test set, its gains are not consistent across years. Similarly, TAI improves performance across both years for explicit queries, but still remains below TPOUR Contriever. In contrast, TPOUR Contriever achieves the best performance across all explicit and implicit settings, improving

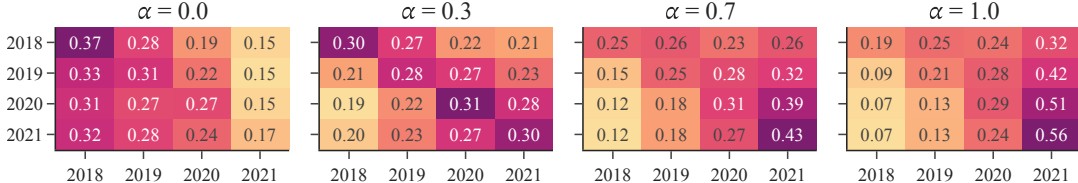

*Figure 3.* Distribution of retrieved document timestamps with time vector interpolation. Heatmaps show the normalized distribution of retrieved document timestamps in years (x-axis) for each test year (y-axis) on SituatedQA. Each heatmap corresponds to a TPOUR Contriever interpolated between retrievers trained on $t_{start}$ = 2018 and $t_{end}$ = 2021, using weights $\alpha$, where 0.0 represents the 2018 and 1.0 represents the 2021 model. Retrieved documents are concentrated around the test year when the interpolation weights align, and shift across intermediate years (2019, 2020) as interpolation value changes, showing temporal alignment in intermediate years.

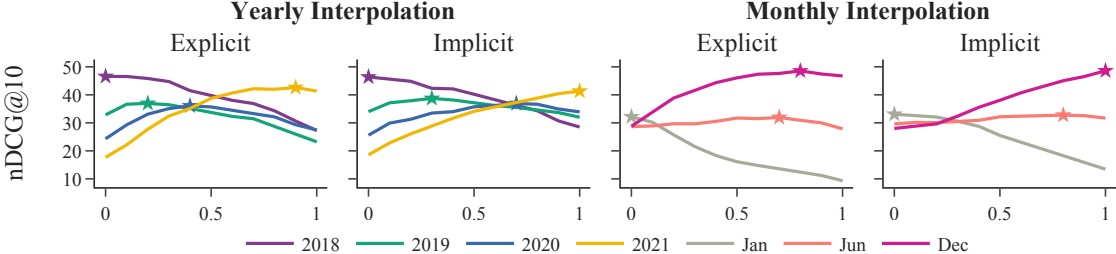

*Figure 4.* Temporal retrieval performance of interpolated TPOUR Contriever. nDCG@10 on **Left**: SituatedQA (Yearly) and **Right**: RealTimeQA (Monthly) using interpolated TPOUR Contriever between $\pi_\theta^{t_{start}}$ and $\pi_\theta^{t_{end}}$ (2018/2021 for SituatedQA, January/December 2023 for RealTimeQA), evaluated with explicit and implicit temporal information in queries. The x-axis indicates the interpolation weight $\alpha$ between 2018 and 2021. Each colored line denotes an evaluation set, and star markers (★) indicate the interpolation achieving peak performance. Peaks aligning with the corresponding time period show temporal generalization across intermediate periods.

*Table 5.* TPOUR Contriever outperforms 72.7× larger Qwen3-Embedding-8B variants with query rewriting (QR) and temporally aware instruction (TAI) on explicit and implicit SituatedQA, showing that large embedding models alone cannot fully resolve temporal misalignment without temporal preference optimization.

| Retriever | 2018 | | 2021 | |
|---|---|---|---|---|
| | N@5 | N@10 | N@5 | N@10 |
| Query with Explicit Temporal Information | | | | |
| Qwen3-Embedding-8B | 30.45 | 33.77 | 35.17 | 37.85 |
| Qwen3-Embedding-8B (QR) | 40.92 | 44.07 | 34.46 | 37.23 |
| Qwen3-Embedding-8B (TAI) | 32.69 | 35.48 | 38.51 | 40.64 |
| TPOUR Contriever | **43.93** | **46.66** | **40.21** | **44.72** |
| Query with Implicit Temporal Information | | | | |
| Qwen3-Embedding-8B | 28.07 | 32.60 | 33.91 | 36.45 |
| Qwen3-Embedding-8B (QR) | 29.12 | 32.84 | 31.77 | 34.48 |
| TPOUR Contriever | **44.11** | **46.59** | **39.40** | **44.72** |

nDCG@5 over the strongest Qwen3-Embedding-8B variant by +3.01 in 2018 and +1.70 in 2021 for explicit queries, and by +14.99 in 2018 and +5.49 in 2021 for implicit queries. Tab. 3 shows interpolated TPOUR Contriever performance. On SituatedQA, interpolated retrievers achieve an average improvement of +13.4 nDCG@5 over the start-year retriever and +10.8 over the end-year retriever, relative to the best interpolation setting. RealTimeQA shows similar trends, with interpolation improving nDCG@5 by +9.0 points on average compared to retrievers trained on fixed January or December snapshots. Importantly, interpolated retrievers

match or outperform retrievers trained directly at the middle time (*i.e.*, $\alpha = 0.5$), demonstrating that interpolation enables continuous generalization across time without explicit retraining. Full results across all years and months are provided in Tab. 12, 13, and 14 in the Appendix.

Fig. 4 illustrates how interpolation enables TPOUR to adapt to continuous time shifts. Retrieval performance peaks when the interpolation weight aligns with the test timestamp. For instance, interpolated TPOUR Contriever achieves peak nDCG@10 on the 2019 (green line) and 2020 (blue line) test sets in SituatedQA when interpolation is around the intermediate period. Similarly, on RealTimeQA, the interpolated retriever peaks on the June test set (orange line). We also conduct an ablation study on the loss weight $\lambda$, which balances semantic and temporal supervision, as shown in Appendix Fig. 10. We find that moderate values of $\lambda$ (0.7–0.85) yield the optimal performance.

### 4.4.3. CAN TEMPORAL AWARENESS REVEAL TIME SENSITIVITY IN GENERAL RETRIEVAL?

To assess whether temporal awareness can provide insights into general retrieval tasks, we evaluate TPOUR on the BEIR benchmark spanning diverse domains and creation years. As shown in Fig. 5 and Appendix Tab. 11, interpolated TPOUR Contriever between 2018 and 2021, along with interpolation values $\alpha$ for 2021, reveal clear trends. Older datasets (*e.g.*, MS MARCO) perform best when $\alpha = 0.0$, while newer

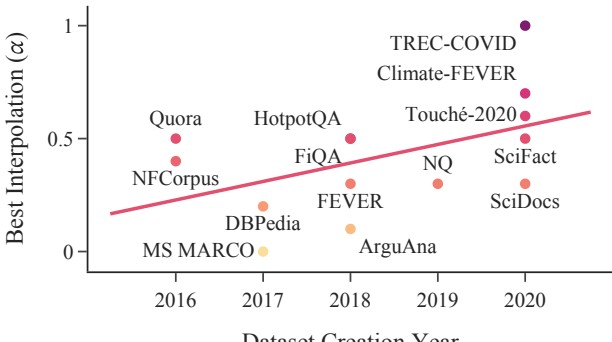

*Figure 5.* Best-performing interpolation $\alpha$ for each BEIR dataset relative to its creation year. Each point denotes a dataset, where $\alpha$ is the interpolation weight for 2021 between TPOUR Contriever (2018) and (2021). The red regression line indicates that datasets prefer retrievers temporally aligned with their publication year. For example, Climate-FEVER (2020) achieves peak performance at $\alpha = 0.7$. Time-sensitive datasets such as TREC-COVID favor higher $\alpha$, whereas less sensitive ones (SciFact, SciDocs) perform well with lower weights. Full results are in Appendix Tab. 11.

datasets (*e.g.*, TREC-COVID and Climate-FEVER) peak when interpolated toward 2021 (*i.e.*, $\alpha = 1.0$). These results show that temporal awareness reveals time sensitivity in retrieval, aligning with dataset years.

We conduct a qualitative case study comparing outputs from Contriever and TPOUR Contriever. As shown in Appendix Tab. 20 and 21, TPOUR Contriever retrieves documents that are both semantically relevant and temporally aligned with the query. For example, given "When did the Golden State Warriors win the Finals as of 2018," TPOUR Contriever returns documents about the 2018 NBA Finals, whereas Contriever retrieves general descriptions of the NBA Finals. Similarly, for "Who has won the most Olympic medals in curling as of 2021," TPOUR Contriever retrieves temporally aligned documents, whereas Contriever returns older ones.

#### 4.4.4. EXTRAPOLATING TO FUTURE TIME PERIODS

TPOUR uses time-vector interpolation to generalize to intermediate time periods without additional training. Extending this idea to future or more recent time periods would further improve its practical utility. Thus, we conduct an analysis of time-vector extrapolation for future time periods. Specifically, we construct an extrapolated retriever by combining three temporally distinct time vectors extracted from TPOUR models trained on the 2018, 2021, and 2022 document dumps. We define the extrapolated TPOUR retriever as $\theta_{\text{future}} = \theta_{\text{base}} + (1 - \alpha)\tau_{t_{2018}} + \alpha(\tau_{t_{2022}} - \tau_{t_{2021}})$, where $\alpha$ controls the extrapolation strength. Here, $\tau_{t_{2018}}$ denotes the base time vector from which extrapolation is performed, while $\tau_{t_{2021}}$ and $\tau_{t_{2022}}$ denote time vectors obtained from later document dumps. The difference vector $\tau_{t_{2022}} - \tau_{t_{2021}}$ captures the temporal direction from an earlier to a later period.

*Table 6.* Results of time vector extrapolation using RealtimeQA (2023, December) test set. The extrapolated model gained using three temporally outdated retrievers (2018/2021/2022) achieves higher performance than temporally outdated checkpoints.

| Model | N@5 | N@10 |
|---|---|---|
| Oracle: TPOUR Contriever (2023) | 42.45 | 46.15 |
| TPOUR Contriever (2018) | 21.45 | 20.13 |
| TPOUR Contriever (2021) | 23.93 | 25.28 |
| TPOUR Contriever (2022) | 27.78 | 27.99 |
| Extrapolated TPOUR ($\alpha = 0.5$) | **30.00** | **30.40** |

By adding this direction to $\tau_{t_{2018}}$, we approximate a future-oriented time vector beyond the observed training periods. Tab. 6 shows time vector extrapolation performance using the RealTimeQA (2023, December) test set. The results show that an appropriate extrapolation strength ($\alpha = 0.5$) enables TPOUR to approximate future time periods more effectively, outperforming the most recent 2022 TPOUR Contriever baseline in N@5 (30.00 vs. 27.78).

## 5. Conclusion and Future Work

We propose TPOUR, a preference-based training method at the embedding level that injects temporal information into unsupervised dense retrievers. By integrating our TRPO into contrastive learning, TPOUR enables retrievers to learn both semantic similarity and temporal preferences from unlabeled data. We show that time-unaware retrievers suffer from temporal misalignment and that training with TRPO improves on temporal retrieval tasks on SituatedQA and RealTimeQA. We further show that time vector interpolation allows TPOUR-trained retrievers to generalize across continuous time periods without retraining. Beyond temporal retrieval, TPOUR retrievers also exhibit temporal preferences on the BEIR benchmark, indicating that temporal modeling benefits both time-sensitive and general retrieval tasks.

We show that TPOUR improves temporal retrieval, and several promising directions remain for future work. (1) Relaxing the requirement for temporally distributed document collections could broaden applicability. (2) Further analysis of temporal grounding could enhance interpretability across implicit and explicit queries, as the benefits of TPOUR are more pronounced in explicit than in implicit setups. (3) We show that temporal alignment relates to general retrieval. Further studies could expand its usability (*e.g.*, appropriate $\alpha$ selection). Our current setup sets $\alpha$ heuristically based on the test-set time (*e.g.*, $\alpha = 0.3$ between 2018 and 2021 retrievers for the 2019 test set). (4) Time vector extrapolation could enable TPOUR-trained retriever to generalize beyond the training period. Our preliminary results (Sec. 4.4.4, Tab. 6) show that TPOUR can be applied to extrapolation. We provide more details on each aspect of our future work and a more detailed analysis in Appendix F.

## Impact Statement

This paper presents a method for improving temporal alignment in unsupervised information retrieval systems. Improved temporal grounding can enhance the reliability of retrieved information. The method uses existing document corpora. As such, it does not directly raise concerns related to privacy or content misuse.

## Acknowledgments

We would like to thank the anonymous reviewers for their helpful questions and comments. This work was partly supported by Institute of Information & communications Technology Planning & Evaluation(IITP) grant funded by the Korea government(MSIT) (RS-2019-II190421, AI Graduate School Support Program(Sungkyunkwan University) & RS-2025-02263169, Detection and Prediction of Emerging and Undiscovered Voice Phishing & RS-2024-00398115 , Research on the reliability and coherence of outcomes produced by Generative AI). This work was supported by the Ministry of Education of the Republic of Korea and the National Research Foundation of Korea (NRF-RS-2025-00523385).

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

# A. Reproducibility Statements

## A.1. System Architecture and Inference Process

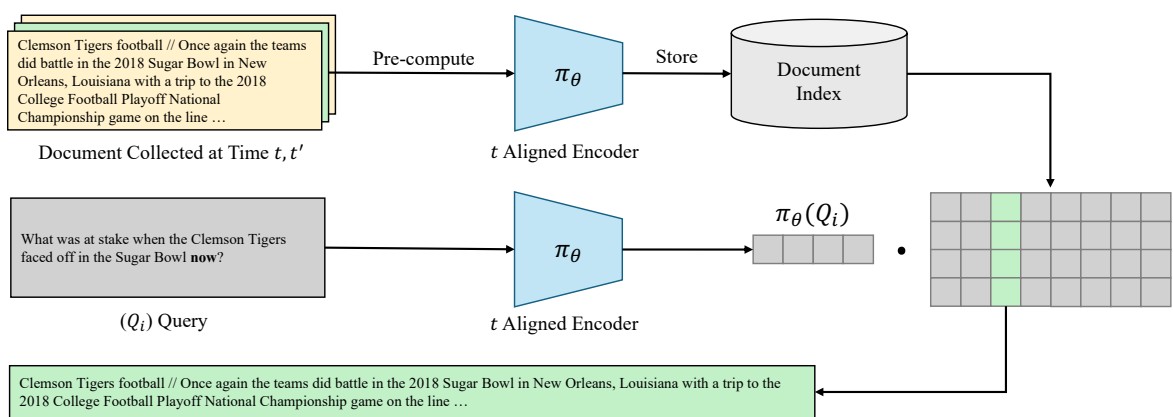

*Figure 6.* An illustration of TPOUR inference. Like standard retrieval, we use the trained encoder $\pi_\theta$ to pre-compute representations for all documents at mixed-timestamps $t$ and $t'$, which are then stored in the document index. At inference, a query $Q_i$ is encoded as $\pi_\theta(Q_i)$, and retrieves the document from the index with the highest similarity to the query. The retrieved document is both semantically relevant and temporally aligned with the query.

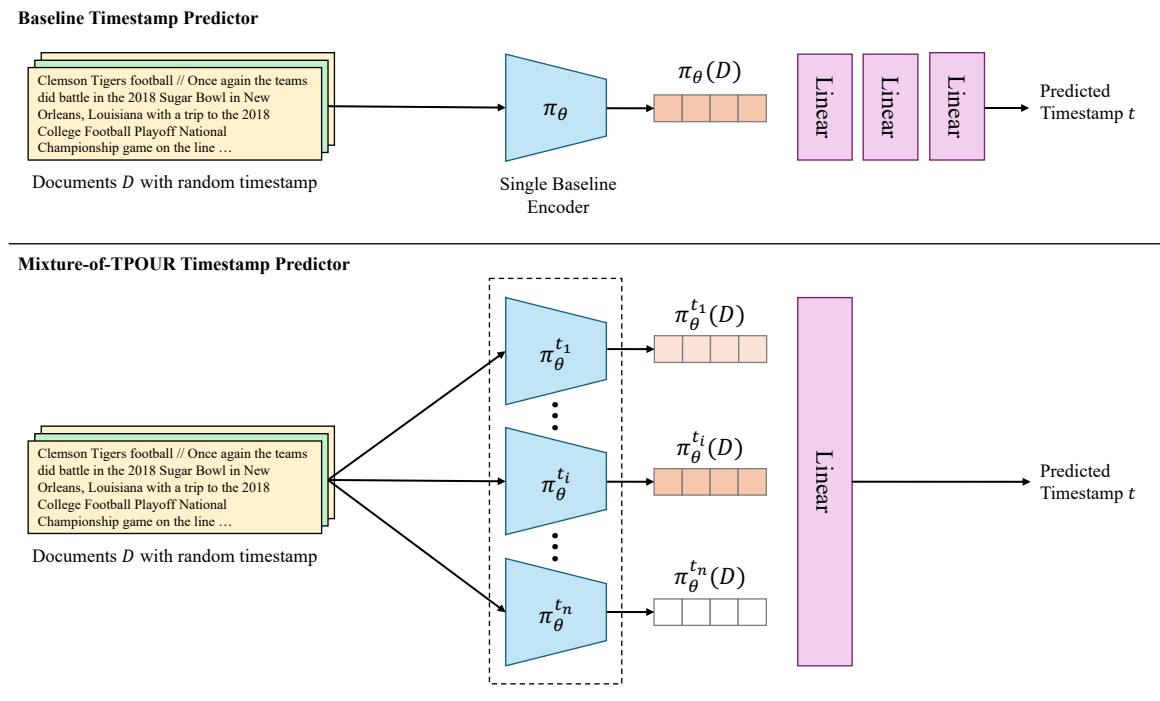

*Figure 7.* An illustration of the Baseline and the mixture-of-TPOUR Timestamp Predictor under a setup where the linear classifier has the same number of parameters. Given a document, the baseline model (upper) uses a single encoder to generate a representation, which is then passed to a linear classifier to predict the timestamp. In contrast, the mixture-of-TPOUR (lower) uses a set of frozen retrievers $\{\pi_\theta^{t_1}, \ldots, \pi_\theta^{t_n}\}$, each specialized for a different time period, to produce temporally-aware embeddings. These are concatenated and fed into a linear classification layer to predict the most likely timestamp. For a fair comparison, we matched the total number of trainable parameters by stacking multiple linear layers in the baseline predictor equal to the number of TPOUR encoders.

## A.2. Training Dataset Construction Procedure

We extract document texts from each dump using Wikiextractor (Attardi, 2015). As summarized in Tab. 7, we first filter out short documents (>50 words), which are mostly hyperlink pages with no content. We then identify overlapping documents across timestamps (Intersection) and retain only those with content changes (Filtered Intersection). Finally, we include timestamp-specific unique documents (Unique) to build the final dataset (Final), ensuring that each timestamp-specific collection contains meaningful temporal differences. Lastly, we remove all documents that appear in the test sets to prevent any data leakage during evaluation. The resulting dataset comprises temporally distinct document collections from each Wikipedia dump, with minimal explicit mentions of the target year. As shown in Tab. 8, fewer than 2.5% of documents contain the target year explicitly within their content.

*Table 7.* Statistics of Wikipedia dumps used for monthly and yearly training & evaluation. (Original) Starting from the full set of documents (>50 words), we filter out those with fewer than 50 words. (Intersection) We then identify overlapping documents across timestamps (Filtered Intersection), further filter for documents that changed between each dump set, and (Unique) add unique documents that are created only at the specific dump set (Final) to obtain the final dataset.

| Dump Set | Original | >50 words | # Docs - Monthly Intersection | Filtered Intersection | Unique | Final |
|---|---|---|---|---|---|---|
| 2023-01-01 | 16,228,228 | 4,876,682 | 4,842,453 | 736,527 | 34,229 | 770,756 |
| 2023-07-01 | 16,505,531 | 4,963,032 | 4,842,453 | 736,527 | 120,579 | 857,106 |
| 2023-12-20 | 16,619,644 | 5,011,040 | 4,842,453 | 736,527 | 168,587 | 905,114 |
| | | | # Docs - Yearly | | | |
| 2018-12-20 | 13,717,022 | 4,021,080 | 3,888,123 | 2,674,468 | 132,957 | 2,807,425 |
| 2021-12-20 | 15,567,219 | 4,706,705 | 3,888,123 | 2,674,468 | 818,582 | 3,493,050 |
| 2023-12-20 | 16,619,644 | 5,011,040 | 3,888,123 | 2,674,468 | 1,122,917 | 3,797,385 |

*Table 8.* Percentage of documents in each Wikipedia dump that contain an explicit mention of the corresponding collection year. As shown, the majority of documents (>97%) do not include lexical references to the target year, reinforcing that TPOUR learns temporal preferences from semantic drift across documents collected at different times, rather than from explicit timestamp information.

| Dump Set | Target Year in Document (%) | Target Year not in Document (%) |
|---|---|---|
| 2018-12-20 | 2.50 | 97.50 |
| 2021-12-20 | 1.89 | 98.01 |
| 2023-12-20 | 1.10 | 98.90 |

## A.3. Training & Evaluation Environment

### A.3.1. TRAINING CONFIGURATION

We fully fine-tune TPOUR using Contriever (Izacard et al., 2022) as the base model (TPOUR Contriever) on a single NVIDIA A100 (80GB) GPU, an AMD EPYC 7763 64-core CPU, and 200GB of memory. The hyperparameters used for TPOUR training are listed in Tab. 9. We use a learning rate of $1\mathrm{e}{-6}$, 4,000 warmup steps, and a MoCo queue of length 131,060. The temporal preference objective is combined with the contrastive loss using $\lambda = 0.925$. We apply token deletion augmentation with a probability of 10%, and chunk input texts to a maximum length of 256 tokens. All normalization options are disabled to preserve the original text form. We use three different random seeds to train TPOUR Contriever.

*Table 9.* Hyperparameters used for training the TPOUR Contriever.

| Hyperparameter | Value |
|---|---|
| Contrastive / TRPO Loss Weight ($\lambda$) | 0.925 |
| Temperature ($T$) | 0.05 |
| Optimizer | AdamW |
| AdamW $\beta_1, \beta_2, \epsilon$ | 0.9, 0.98, 1e$-$6 |
| Learning Rate | 1e$-$6 |
| Scheduler / Warmup Steps | Linear / 4000 |
| Batch Size | 10 |
| MoCo Queue Size | 131,060 |
| Momentum ($m$) | 0.9999 |
| Projection Size | 768 |
| Dropout Rate | 0.1 |
| Chunk Length | 256 |
| Text Augmentation | Deletion (prob = 0.1) |
| Normalization (Query / Doc / Text) | False / False / False |
| Training Steps | 100,000 |
| Data Augmentation (Random Cropping / Delete) | False / True (10%) |

### A.3.2. COMPUTATIONAL COST

TPOUR Contriever is based on BERT-base-uncased (110M parameters, 440MB). For interpolation or mixture-of-TPOUR experiments, we train two TPOUR Contrievers (2018 and 2021), each taking 4.5 GPU hours on a single A100. These are then interpolated to produce 10 time-specific models, resulting in a total storage of 4.4GB. For the mixture-of-TPOUR predictor, the training requires 16 GPU hours (10 hours for the baseline). The predictor uses 0.12M trainable parameters, of about 600KB in size, while all TPOUR retrievers remain frozen.

## B. Theoretical Basis of TPOUR

### B.1. Temporal Retrieval Preference Optimization

Direct Preference Optimization (DPO) (Rafailov et al., 2023) forms a preference pair given a prompt ($x$), preferred ($y^w$) and less preferred response ($y^l$) as ($x, y^w, y^l$). Like DPO, Temporal Retrieval Preference Optimization (TRPO) forms ($Q, D^t, D^{t'}$) as a pairwise preference pair over a timestamped document corpus given a query $Q$, time-aligned document ($D^t$), and unaligned document ($D^{t'}$). The goal of TRPO is to prefer temporally aligned document ($D^t$) over the misaligned ($D^{t'}$) given a query ($Q$) and minimizing the TRPO loss function ($\mathcal{L}_{\text{TRPO}}$) in Eq. 4.

$\mathcal{L}_{\text{TRPO}}$ is based on the Bradley-Terry model. While DPO aligns model score with human-labeled preference, TRPO aligns score with temporal relevance with an implicit signal derived from corpus-level differences (preferring $D^t$ over $D^{t'}$). In this view, TRPO requires working under the following three conditions.

1. **Temporal preference margin.** There must be a certain temporal preference gap (*i.e.*, margin) between aligned and misaligned document $\mathbf{E}[S(Q, D^t) - S(Q, D^{t'})] > \delta$ when $t' \neq t$ where $\delta$ is a minimum gap required. If the actual document update with temporal change is too small relative to noise, TRPO learning could be unstable. To handle this issue, we comprise a temporally distinct document collection by filtering the dataset in Appendix A.2.

2. **Similar semantic across corpora.** Aligned and misaligned temporal corpora should cover a similar set of topics, so semantic similarity may remain high and the only difference is the timestamp and the document content at that timestamp.

3. **Model capacity.** Encoder should have sufficient capacity to represent latent temporal signal as well as semantic similarity.

Under these conditions, TRPO encourages the model to rank temporally aligned documents higher. The resulting scoring function $S_\theta$ is expected to approximate one that reflects temporal alignment between query and document. This mirrors the theoretical guarantees for DPO by replacing "generation quality" with "temporal relevance" as the underlying reward (Wang et al., 2024a; Xiong et al., 2024). To sum up, TRPO is a preference alignment variant, where preferences are defined by temporal grounding between a versioned corpus. This generalizes preference learning to the temporal dimension.

## B.2. Time Vector Interpolation

The assumption that time vectors (*i.e.*, model parameters trained on temporally adjacent corpora) are close in weight space is supported both empirically in Fig. 3, where retrieved documents change smoothly across interpolated models, showing continuity in the learned representation space. Theoretically, time vector interpolation is supported in two parts.

**Distributional similarity leads to weight-space proximity.** Let $P_t$ and $P_{t'}$ be training distributions at time $t$ and $t'$. If $P_t \approx P_{t'}$ (*e.g.*, under low $\text{KL}(P_t|P_{t'})$), then under gradient descent, the learned parameters $\theta_t \approx \theta_{t'}$ will be nearby in weight space. The idea is formalized by (Goodfellow & Vinyals, 2015) and aligns with our setup, where temporally adjacent corpora (*e.g.*, 2018 vs. 2019) are close in weight space. For adjacent periods, only temporal preferences differ, while the training data come from similar Wikipedia distributions.

**Interpolation preserves generalization.** Prior work has shown that models trained on related tasks or distributions often lie in connected regions of the loss landscape (Izmailov et al., 2018; Rame et al., 2023). In our setting, $\theta_t$ and $\theta_{t'}$ are trained on temporally adjacent corpora (*e.g.*, 2018 vs. 2019), which tend to share topical and linguistic structures, yielding a time vector $\tau$. As shown by (Izmailov et al., 2018), linearly interpolating such weight vectors, $\alpha\tau_t + (1 - \alpha)\tau_{t'}$, often produces low-loss solutions if the endpoints lie in a shared basin. This smoothness in weight space supports generalization and has been used in practice via stochastic weight averaging (SWA). Also, (Rame et al., 2023) shows that interpolating across models trained on diverse but related domains can produce generalizable models that outperform the individual components. Analogously, we treat time as an axis of distributional change, and our interpolation procedure leverages this continuity to produce retrievers that generalize to intermediate periods.

# C. Related Work in Information Retrieval

## C.1. Early Work on Temporal Alignment in Information Retrieval

(Berberich et al., 2010) explored the inherent uncertainty of temporal expressions and proposed representing them as tuples, integrating this representation into a probabilistic language modeling framework for information retrieval. (Jatowt et al., 2005) proposed a re-ranking method that utilizes archived web snapshots to prioritize documents based on content freshness and relevance. They also introduced the concept of document focus time, which refers to the temporal period indicated by the document content and is distinct from its creation time. Additionally, they proposed a method to automatically estimate this temporal reference using large news collections and external knowledge bases (Jatowt et al., 2013). (Kanhabua & Nørvåg, 2010) developed methods for determining the time of implicit temporal queries by leveraging temporal language models trained on timestamped corpora. They further proposed the first machine learning framework capable of automatically selecting the most effective temporal ranking strategy for a given query (Kanhabua et al., 2012).

## C.2. Baseline Models

**DPR (Dense Passage Retrieval)** (Karpukhin et al., 2020) is a supervised dense retriever trained on query-passage pairs using a bi-encoder architecture. It optimizes retrieval by maximizing similarity between queries and relevant passages while minimizing similarity to negative samples. DPR is trained using hard negatives from BM25 to improve retrieval quality.

**Contriever** (Izacard et al., 2022) is a self-supervised dense retriever trained with contrastive learning, removing the need for labeled query-document pairs. It constructs high-quality negative samples using a momentum encoder, enabling scalable pretraining on large unlabeled corpora.

**REALM (Retrieval-Augmented Language Model)** (Guu et al., 2020) jointly trains a dense retriever and a language model in an end-to-end manner. During pretraining, the retriever is updated to select relevant documents that improve language model performance. This integration enables the model to dynamically leverage external knowledge, making it particularly effective for knowledge-intensive NLP tasks such as open-domain QA.

**SimCSE** (Gao et al., 2021) is a sentence embedding model trained using contrastive learning in both supervised and unsupervised settings. The unsupervised variant leverages dropout as noise, while the supervised variant uses natural language inference (NLI) data. Though not originally intended for retrieval, SimCSE embeddings can be used for dense retrieval by comparing query and document representations in a shared semantic space.

**Temporal Language Modeling** (Berberich et al., 2010) is a retrieval framework that integrates temporal expressions into language models by modeling their inherent uncertainty. The proposed uncertainty-aware model represents temporal

expressions as interval distributions and measures temporal relevance via overlap between query and document intervals.

**Temporal Contrastive** is a temporally-aware contrastive baseline that augments the standard contrastive retrieval objective with temporal supervision. We include this baseline to examine whether temporal alignment can be obtained by directly constructing time-based positive and negative pairs, without preference-based optimization. For each query, temporally aligned documents are treated as positives and temporally misaligned documents as negatives, yielding $\mathcal{L}_{\text{TempCE}}$, which encourages higher similarity between the query and documents that better match the target time. The final objective is $\mathcal{L}_{\text{total}} = \lambda \mathcal{L}_{\text{CE}} + (1 - \lambda)\mathcal{L}_{\text{TempCE}}$, where $\mathcal{L}_{\text{CE}}$ models semantic relevance and $\mathcal{L}_{\text{TempCE}}$ models temporal alignment.

**TimeR**[4] (Qian et al., 2024) is a retrieval-augmented generation framework for temporal knowledge graph question answering. It includes a time-aware dense retriever trained with contrastive learning to capture both semantic and temporal constraints. In our experiments, we use its retriever component for evaluation.

**Nomic Embed v2 MoE** (Nussbaum & Duderstadt, 2025) is a sparse Mixture-of-Experts (MoE) embedding model developed for efficient and scalable dense retrieval. It activates a small subset of expert networks per input, balancing high capacity with low inference cost. Trained using hard negative mining and consistency filtering, it achieves competitive retrieval performance compared to fully dense models. As a general-purpose model, it is open-sourced and designed to perform well across various domains and tasks without extensive fine-tuning.

**Qwen3-Embedding-8B** (Zhang et al., 2025) is a large-scale embedding model with 8B parameters, built upon Qwen3 (Yang et al., 2025). It supports diverse embedding and reranking tasks across multiple domains and languages. We consider three retrieval setups. (1) Naive retrieval: As in conventional retrieval methods, we directly use the original query to retrieve relevant documents. (2) Query rewriting retrieval: This method uses a large language model to rewrite the query to make the temporal intent more explicit. Specifically, we use GPT-OSS-20B (OpenAI et al., 2025) as a frozen rewriter, following prior work (Ma et al., 2023). (3) time-aware instruction retrieval: Since Qwen3-Embedding-8B supports instruction-conditioned embeddings, we apply TAI to queries with explicit temporal information, where the target time can be directly encoded in the instruction prompt shown in Tab. 10.

*Table 10.* Instruction-aware prompting template for temporal retrieval with Qwen-3-Embedding-8B. {QUERY} denotes a placeholder.

| **Prompt Content** |
| --- |
| You are a retrieval system that selects documents relevant to a query. |
| Temporal requirement: |
| - Preference: prioritize documents whose content reflects knowledge available at that time. |
| - Avoid documents containing information published after the target time unless explicitly requested. |
| |
| Instructions: |
| 1. Identify the temporal intent of the query. |
| 2. Filter or downweight documents that violate the temporal constraint. |
| 3. Rank documents by both semantic relevance and temporal alignment. |
| 4. Prefer documents whose timestamps are closest to, but not exceeding, the target time. |
| Query: {QUERY} |

# D. Additional Experimental Results & Analysis

## D.1. Full Results on BEIR Benchmark

*Table 11.* Retrieval performance (nDCG@10) on the BEIR benchmark, with dataset publication years shown below each dataset name. Each benchmark exhibits specific temporal preferences that mostly align with its creation date, suggesting that TPOUR can improve general retrieval performance by adapting to the temporal characteristics of different datasets.

| Interpolation 2018 | 2021 | 2021 | 2023 | MS MARCO 2017 | TREC-COVID 2020 | NFCorpus 2016 | NQ 2019 | HotpotQA 2018 | FiQA 2018 | ArguAna 2018 | Touché-2020 2020 | Quora 2016 | DBPedia 2017 | SciDocs 2020 | FEVER 2018 | Climate-FEVER 2020 | SciFact 2020 |
|---|---|---|---|---|---|---|---|---|---|---|---|---|---|---|---|---|---|
| 1 | 0 | | | 45.56 | 21.44 | 25.52 | 23.90 | 40.88 | 19.22 | 43.65 | 9.39 | 81.98 | 30.11 | 13.51 | 45.63 | 19.75 | 60.68 |
| 0.9 | 0.1 | | | 45.30 | 22.57 | 25.98 | 24.23 | 41.52 | 19.75 | 43.69 | 9.83 | 82.01 | 30.42 | 13.74 | 46.37 | 20.49 | 60.95 |
| 0.8 | 0.2 | | | 45.14 | 22.60 | 26.36 | 24.52 | 42.00 | 20.31 | 43.57 | 10.44 | 82.15 | 30.67 | 13.86 | 46.72 | 21.08 | 61.12 |
| 0.7 | 0.3 | | | 44.98 | 23.11 | 26.91 | 24.82 | 42.41 | 20.51 | 43.56 | 10.80 | 82.15 | 30.62 | 13.88 | 46.85 | 21.62 | 61.34 |
| 0.6 | 0.4 | | | 44.33 | 23.42 | 27.18 | 24.82 | 42.60 | 20.54 | 43.37 | 10.92 | 82.22 | 30.52 | 13.82 | 46.69 | 22.11 | 61.57 |
| 0.5 | 0.5 | | | 43.84 | 24.16 | 27.17 | 24.80 | 42.61 | 20.56 | 43.52 | 10.91 | 82.23 | 30.35 | 13.77 | 46.19 | 22.59 | 61.73 |
| 0.4 | 0.6 | | | 43.40 | 24.98 | 27.03 | 24.51 | 42.40 | 20.25 | 43.53 | 10.95 | 82.21 | 29.74 | 13.61 | 45.33 | 22.62 | 61.26 |
| 0.3 | 0.7 | | | 42.78 | 25.07 | 26.90 | 24.21 | 41.92 | 20.16 | 43.48 | 10.79 | 82.14 | 29.31 | 13.39 | 44.07 | 22.68 | 61.26 |
| 0.2 | 0.8 | | | 41.26 | 25.74 | 26.64 | 23.82 | 41.30 | 19.78 | 43.38 | 10.85 | 82.00 | 28.74 | 13.08 | 42.50 | 22.68 | 60.56 |
| 0.1 | 0.9 | | | 40.90 | 26.35 | 26.27 | 23.08 | 40.38 | 19.15 | 42.99 | 10.29 | 81.84 | 28.10 | 12.81 | 40.55 | 22.44 | 59.58 |
| 0 | 1 | | | 39.95 | 27.08 | 25.78 | 22.58 | 39.27 | 18.29 | 42.69 | 9.99 | 81.67 | 27.16 | 12.43 | 38.49 | 22.13 | 59.35 |
| | | 1 | 0 | 39.95 | 27.08 | 25.78 | 22.58 | 39.27 | 18.29 | 42.69 | 9.99 | 81.67 | 27.16 | 12.43 | 38.49 | 22.13 | 59.35 |
| | | 0.9 | 0.1 | 40.78 | 26.95 | 26.40 | 22.96 | 40.06 | 18.71 | 42.80 | 10.06 | 81.85 | 27.84 | 12.81 | 39.88 | 22.15 | 59.59 |
| | | 0.8 | 0.2 | 41.84 | 27.03 | 26.72 | 23.19 | 40.59 | 19.13 | 42.96 | 10.28 | 81.94 | 28.36 | 13.09 | 40.86 | 22.17 | 59.99 |
| | | 0.7 | 0.3 | 41.89 | 28.13 | 26.96 | 23.40 | 40.90 | 19.38 | 42.89 | 10.74 | 81.99 | 28.54 | 13.20 | 41.72 | 22.14 | 60.53 |
| | | 0.6 | 0.4 | 42.25 | 28.72 | 26.78 | 23.48 | 41.05 | 19.51 | 42.92 | 10.92 | 81.98 | 28.58 | 13.38 | 42.23 | 21.89 | 60.63 |
| | | 0.5 | 0.5 | 42.43 | 28.93 | 26.75 | 23.48 | 41.12 | 19.40 | 43.09 | 10.99 | 81.99 | 28.71 | 13.25 | 42.17 | 21.62 | 60.55 |
| | | 0.4 | 0.6 | 42.26 | 28.12 | 26.43 | 23.25 | 40.97 | 19.01 | 43.12 | 11.41 | 81.98 | 28.69 | 13.18 | 41.98 | 21.16 | 60.49 |
| | | 0.3 | 0.7 | 42.43 | 28.51 | 25.96 | 22.89 | 40.58 | 18.77 | 43.11 | 10.92 | 81.92 | 28.23 | 13.08 | 41.44 | 20.38 | 60.20 |
| | | 0.2 | 0.8 | 42.63 | 28.91 | 25.12 | 22.56 | 40.11 | 18.58 | 43.26 | 10.61 | 81.71 | 27.67 | 12.82 | 40.79 | 19.65 | 59.27 |
| | | 0.1 | 0.9 | 42.21 | 28.80 | 24.38 | 22.20 | 39.50 | 18.06 | 43.13 | 10.09 | 81.44 | 27.38 | 12.44 | 39.98 | 18.79 | 58.73 |
| | | 0 | 1 | 41.31 | 29.11 | 23.54 | 21.68 | 38.60 | 17.35 | 43.18 | 10.38 | 81.14 | 27.04 | 12.18 | 38.90 | 18.00 | 57.81 |

## D.2. Full Results on Interpolation

*Table 12.* TPOUR yearly transition in performance with interpolation on SituatedQA. The color saturation indicates the relative performance, with darker green representing higher scores within each column. The table shows the impact of time vector interpolation on retrieval performance across different time periods, where the highest scores are achieved at their corresponding evaluation times. Gradual changes in performance are observed as the interpolation values shift.

| Interpolation | | nDCG@5 | | | | nDCG@10 | | | | Recall@5 | | | | Recall@10 | | | |
| --- | --- | --- | --- | --- | --- | --- | --- | --- | --- | --- | --- | --- | --- | --- | --- | --- | --- |
| 2018 | 2021 | 2018 | 2019 | 2020 | 2021 | 2018 | 2019 | 2020 | 2021 | 2018 | 2019 | 2020 | 2021 | 2018 | 2019 | 2020 | 2021 |
| 1 | 0 | 44.10 | 30.83 | 21.66 | 14.00 | 46.57 | 32.84 | 24.30 | 17.68 | 34.46 | 26.04 | 21.28 | 12.84 | 46.18 | 34.69 | 29.89 | 20.38 |
| 0.9 | 0.1 | 43.38 | 33.57 | 26.19 | 18.86 | 46.55 | 36.64 | 29.28 | 22.15 | 34.66 | 28.31 | 26.15 | 17.64 | 47.04 | 39.38 | 36.09 | 25.85 |
| 0.8 | 0.2 | 42.39 | 33.92 | 29.75 | 24.14 | 45.83 | 36.95 | 33.10 | 27.71 | 35.28 | 30.44 | 28.35 | 23.04 | 47.05 | 42.10 | 39.48 | 33.31 |
| 0.7 | 0.3 | 40.72 | 32.84 | 31.14 | 28.92 | 44.73 | 36.45 | 35.07 | 32.42 | 34.28 | 29.72 | 30.72 | 26.82 | 46.95 | 41.68 | 43.29 | 37.43 |
| 0.6 | 0.4 | 37.93 | 30.94 | 32.61 | 31.29 | 41.49 | 35.13 | 36.07 | 35.12 | 33.72 | 29.20 | 32.14 | 27.46 | 45.92 | 42.10 | 43.38 | 39.55 |
| 0.5 | 0.5 | 35.79 | 28.88 | 32.53 | 35.63 | 39.71 | 33.83 | 35.67 | 38.74 | 33.00 | 28.46 | 31.67 | 29.25 | 46.01 | 42.41 | 42.83 | 40.85 |
| 0.4 | 0.6 | 34.04 | 27.55 | 31.41 | 38.52 | 38.04 | 32.29 | 34.48 | 40.66 | 32.21 | 27.68 | 31.92 | 30.43 | 45.21 | 40.56 | 41.74 | 41.39 |
| 0.3 | 0.7 | 32.24 | 26.77 | 29.67 | 39.80 | 36.89 | 31.47 | 33.19 | 42.16 | 29.87 | 26.79 | 30.06 | 30.09 | 44.54 | 39.45 | 41.08 | 41.56 |
| 0.2 | 0.8 | 28.79 | 24.57 | 28.18 | 40.58 | 34.29 | 28.83 | 32.12 | 41.98 | 26.20 | 25.09 | 26.71 | 30.33 | 42.32 | 37.09 | 39.52 | 39.78 |
| 0.1 | 0.9 | 25.67 | 21.74 | 26.05 | 40.84 | 30.64 | 26.28 | 29.32 | 42.60 | 23.78 | 20.87 | 24.75 | 28.77 | 37.52 | 33.83 | 35.28 | 39.55 |
| 0 | 1 | 22.61 | 19.76 | 24.62 | 40.83 | 27.25 | 23.22 | 27.44 | 41.34 | 22.37 | 19.10 | 22.84 | 27.05 | 33.17 | 29.15 | 32.99 | 36.99 |

*Table 13.* Yearly transition in TPOUR performance with interpolation on SituatedQA, when time information is given implicitly in the query.

| Interpolation | | nDCG@5 | | | | nDCG@10 | | | | Recall@5 | | | | Recall@10 | | | |
| --- | --- | --- | --- | --- | --- | --- | --- | --- | --- | --- | --- | --- | --- | --- | --- | --- | --- |
| 2018 | 2021 | 2018 | 2019 | 2020 | 2021 | 2018 | 2019 | 2020 | 2021 | 2018 | 2019 | 2020 | 2021 | 2018 | 2019 | 2020 | 2021 |
| 1 | 0 | 44.63 | 30.78 | 22.14 | 14.83 | 46.37 | 33.99 | 25.57 | 18.57 | 36.15 | 25.66 | 21.26 | 13.08 | 46.70 | 36.90 | 31.28 | 22.67 |
| 0.9 | 0.1 | 42.69 | 34.04 | 26.57 | 18.68 | 45.48 | 37.15 | 29.90 | 22.82 | 36.22 | 29.67 | 27.16 | 17.48 | 47.08 | 40.56 | 37.04 | 27.51 |
| 0.8 | 0.2 | 40.42 | 34.31 | 27.92 | 21.79 | 44.84 | 37.89 | 31.27 | 26.05 | 33.97 | 30.89 | 28.54 | 21.23 | 48.03 | 42.76 | 37.98 | 32.49 |
| 0.7 | 0.3 | 38.64 | 34.62 | 29.94 | 25.29 | 42.32 | 38.73 | 33.48 | 28.99 | 34.61 | 31.60 | 30.68 | 24.61 | 47.04 | 43.95 | 41.18 | 35.29 |
| 0.6 | 0.4 | 37.62 | 34.65 | 29.85 | 27.93 | 42.12 | 38.15 | 33.99 | 31.65 | 34.45 | 32.12 | 31.15 | 26.16 | 47.80 | 43.66 | 42.21 | 37.26 |
| 0.5 | 0.5 | 35.14 | 33.81 | 31.98 | 30.02 | 40.20 | 37.10 | 35.71 | 34.11 | 32.46 | 32.96 | 32.22 | 27.56 | 47.17 | 43.41 | 43.53 | 39.06 |
| 0.4 | 0.6 | 33.20 | 33.21 | 32.93 | 31.98 | 38.09 | 36.18 | 36.21 | 35.62 | 31.64 | 33.01 | 33.47 | 28.58 | 45.80 | 43.08 | 43.46 | 39.34 |
| 0.3 | 0.7 | 32.24 | 32.06 | 33.14 | 34.36 | 36.03 | 35.50 | 36.90 | 37.17 | 31.34 | 31.86 | 34.21 | 29.36 | 42.54 | 43.32 | 44.35 | 39.54 |
| 0.2 | 0.8 | 28.51 | 30.90 | 32.57 | 36.26 | 34.28 | 34.72 | 36.63 | 38.79 | 26.69 | 30.47 | 33.11 | 30.96 | 42.13 | 42.98 | 45.06 | 40.19 |
| 0.1 | 0.9 | 26.15 | 29.68 | 30.31 | 38.47 | 30.68 | 33.66 | 34.94 | 40.41 | 25.75 | 28.39 | 32.82 | 30.82 | 37.19 | 41.15 | 44.41 | 40.38 |
| 0 | 1 | 23.89 | 28.28 | 29.75 | 38.40 | 28.49 | 31.99 | 33.91 | 41.33 | 22.53 | 27.02 | 31.90 | 29.76 | 35.47 | 39.15 | 42.58 | 42.24 |

*Table 14.* TPOUR monthly performance with interpolation on RealTimeQA, when time information is given explicitly (left) or implicitly (right) in the query. Results show temporal alignment in January (Jan), June (Jun), and December (Dec).

| Interpolation | | nDCG@5 | | | nDCG@10 | | | nDCG@5 | | | nDCG@10 | | |
| --- | --- | --- | --- | --- | --- | --- | --- | --- | --- | --- | --- | --- | --- |
| | | Test Month (Explicit) | | | | | | Test Month (Implicit) | | | | | |
| Jan | Dec | Jan | Jun | Dec | Jan | Jun | Dec | Jan | Jun | Dec | Jan | Jun | Dec |
| 1 | 0 | 32.08 | 29.41 | 29.36 | 32.18 | 28.64 | 28.78 | 32.82 | 29.45 | 27.78 | 33.07 | 29.61 | 28.00 |
| 0.9 | 0.1 | 28.80 | 28.67 | 34.36 | 30.19 | 28.89 | 33.77 | 33.53 | 30.01 | 30.13 | 32.46 | 30.15 | 28.80 |
| 0.8 | 0.2 | 25.26 | 29.98 | 40.20 | 25.73 | 29.67 | 38.85 | 33.55 | 29.88 | 31.59 | 32.07 | 30.08 | 29.62 |
| 0.7 | 0.3 | 21.96 | 30.01 | 44.43 | 21.54 | 29.64 | 41.58 | 32.03 | 30.19 | 33.64 | 30.56 | 30.34 | 32.39 |
| 0.6 | 0.4 | 17.58 | 30.79 | 47.23 | 18.33 | 30.51 | 44.34 | 29.30 | 29.91 | 37.83 | 28.76 | 30.90 | 35.53 |
| 0.5 | 0.5 | 16.00 | 32.39 | 49.87 | 16.09 | 31.72 | 46.06 | 25.32 | 30.68 | 39.72 | 25.50 | 32.17 | 38.05 |
| 0.4 | 0.6 | 14.41 | 32.88 | 49.99 | 14.78 | 31.49 | 47.36 | 20.19 | 31.16 | 42.67 | 23.12 | 32.34 | 40.73 |
| 0.3 | 0.7 | 12.95 | 31.77 | 49.94 | 13.65 | 31.94 | 47.63 | 17.20 | 32.02 | 45.12 | 20.86 | 32.69 | 42.90 |
| 0.2 | 0.8 | 10.86 | 31.80 | 50.98 | 12.53 | 30.90 | 48.54 | 15.24 | 33.57 | 48.10 | 18.55 | 32.76 | 45.03 |
| 0.1 | 0.9 | 9.82 | 30.98 | 51.13 | 11.21 | 29.99 | 47.44 | 13.07 | 34.04 | 50.80 | 15.75 | 32.55 | 46.54 |
| 0 | 1 | 8.41 | 29.57 | 49.98 | 9.33 | 27.84 | 46.75 | 11.80 | 33.47 | 53.12 | 13.45 | 31.68 | 48.59 |

## D.3. Timestamp Distribution of Retrieved Documents

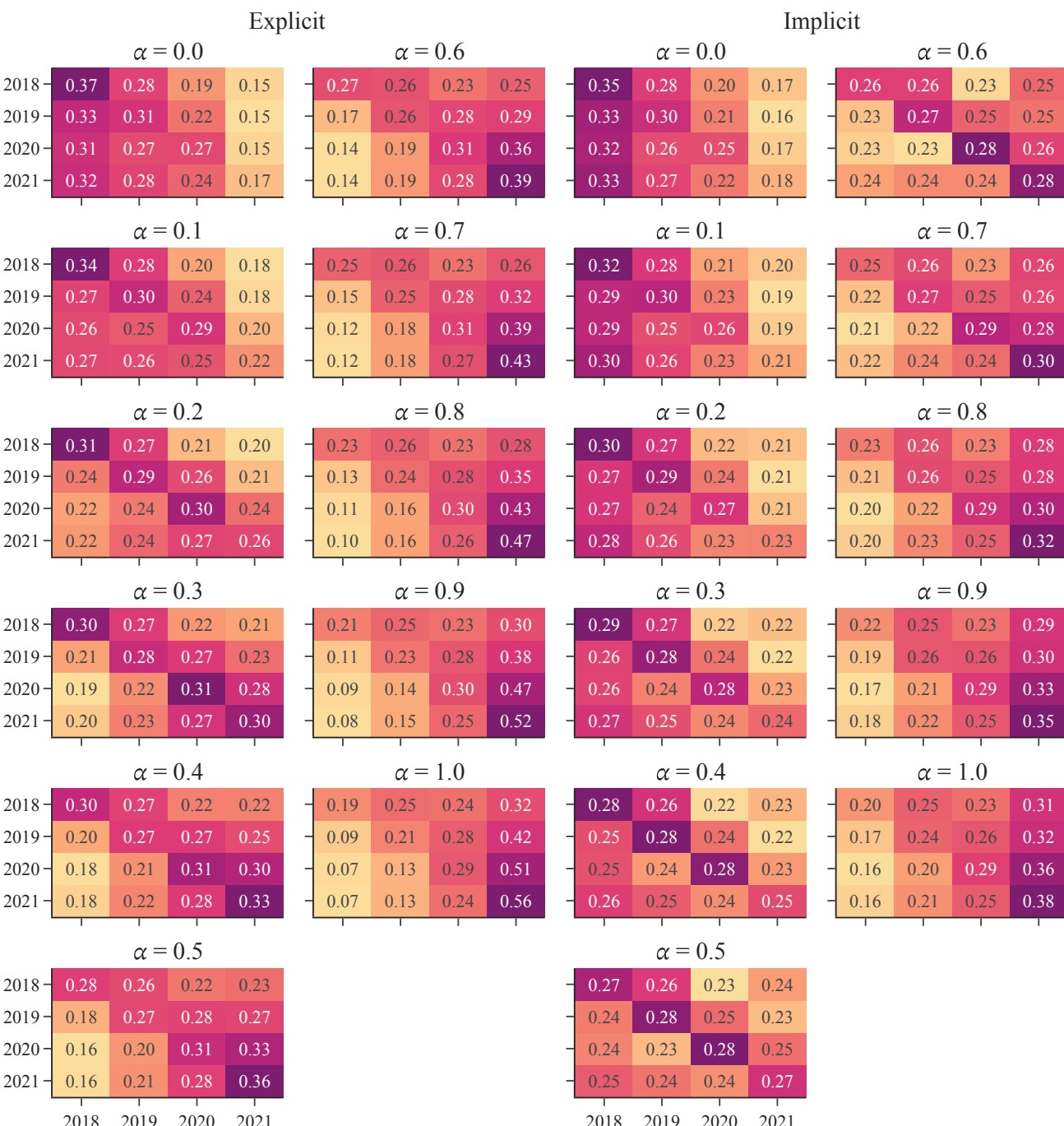

*Figure 8.* Normalized count of retrieved documents per year (X-axis) given the test set year (Y-axis) on SituatedQA, with queries containing explicit (Explicit) or implicit (Implicit) temporal information, when interpolated between 2018 ($\alpha = 0.0$) and 2021 ($\alpha = 1.0$).

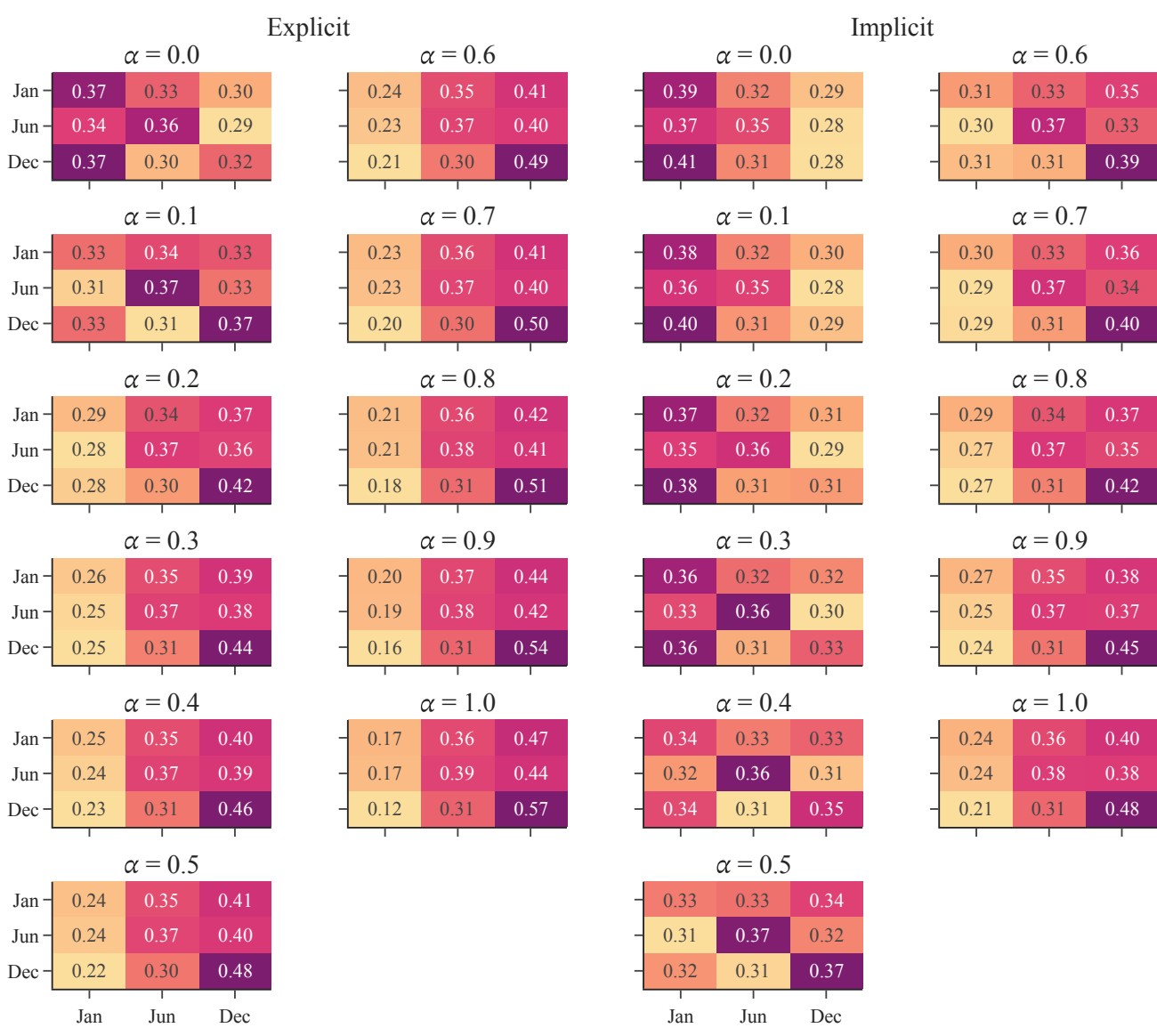

*Figure 9.* Normalized count of retrieved documents per year (X-axis) given the test set year (Y-axis) on RealTimeQA, with queries containing explicit (Explicit) or implicit (Implicit) temporal information, when interpolated between January ($\alpha = 0.0$) and December ($\alpha = 1.0$).

### D.4. Lambda Interpolation

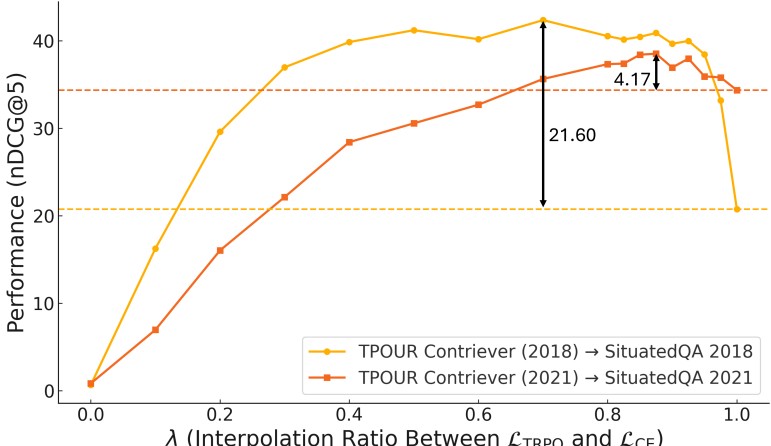

*Figure 10.* Ablation of $\lambda$, the interpolation ratio between $\mathcal{L}_{\text{TRPO}}$ ($\lambda = 0.0$) $\mathcal{L}_{\text{CE}}$ ($\lambda = 1.0$), for TPOUR Contriever 2018 and 2021, evaluated on SituatedQA 2018 and 2021 respectively. Performance improves significantly with moderate $\lambda$ values, showing that combining semantic and temporal supervision is more effective than relying solely on either. Dashed lines at $\lambda = 1.0$ indicate performance using contrastive-only training. Vertical arrows show the performance gap compared to TPOUR 's peak setting for each year.

### D.5. Queue Size Ablation

*Table 15.* Effect of contrastive queue size on temporal retrieval performance for TPOUR-trained retrievers. We vary the queue size used in contrastive training from 100 to 256k while keeping all other training settings fixed. Performance first improves as the queue grows, indicating that a larger set of in-batch negatives helps the model learn stronger temporal preference signals. The optimal queue size is between 4k (50.20) to 16k (44.32).

| Model \ Queue Size | 100 | 500 | 1k | 2k | 4k | 16k | 64k | 128k | 256k |
|---|---|---|---|---|---|---|---|---|---|
| TPOUR Contriever (2018) | 46.87 | 48.56 | 48.88 | 50.01 | **50.20** | 49.36 | 47.78 | 47.38 | 46.96 |
| TPOUR Contriever (2021) | 37.26 | 39.97 | 40.19 | 41.14 | 41.83 | **44.32** | 42.91 | 41.99 | 41.15 |

### D.6. Retrieved Document-Year Distribution Over Training

To verify that TPOUR learns to distinguish content updates over time (beyond matching explicit temporal markers), we track how the *distribution of retrieved document years* changes throughout training. Concretely, at several checkpoints (0k–100k steps), we retrieve documents for a fixed evaluation set and compute the fraction of retrieved documents belonging to each snapshot year (normalized so each column sums to 100%). If the model learns temporal alignment from implicit semantic shifts across versions, the retrieved-year distribution should progressively concentrate around the target snapshot time.

Across training, the retrieved-year distribution shifts toward the snapshot time each model is trained to prefer. TPOUR Contriever (2018) increases the retrieved 2018 documents (25.8→31.0) while decreasing later years, whereas TPOUR Contriever (2021) increasingly concentrates on 2021 documents (37.2→42.1) while reducing earlier years.

*Table 16.* Retrieved document-year distribution (%, normalized) over training steps for TPOUR Contriever (2018).

| Doc Year \ Step | 0k | 20k | 40k | 60k | 80k | 100k |
|---|---|---|---|---|---|---|
| 2018 | 25.8 | 29.9 | 30.6 | 30.8 | 30.9 | 31.0 |
| 2019 | 24.8 | 26.0 | 26.1 | 26.2 | 26.2 | 26.2 |
| 2020 | 23.6 | 22.1 | 21.9 | 21.9 | 21.8 | 21.4 |
| 2021 | 25.7 | 21.9 | 21.3 | 21.1 | 21.1 | 21.4 |

*Table 17.* Retrieved document-year distribution (%, normalized) over training steps for TPOUR Contriever (2021).

| Doc Year \ Step | 0k | 20k | 40k | 60k | 80k | 100k |
|---|---|---|---|---|---|---|
| 2018 | 17.1 | 16.8 | 16.7 | 15.3 | 14.8 | 14.0 |
| 2019 | 19.5 | 20.4 | 19.7 | 18.8 | 18.4 | 18.0 |
| 2020 | 26.3 | 26.3 | 26.3 | 26.2 | 25.7 | 25.8 |
| 2021 | 37.2 | 36.5 | 37.2 | 39.8 | 41.1 | 42.1 |

## D.7. Seasonal Preference of TPOUR-trained Retriever

We analyze preference on temporal patterns using TPOUR. While TPOUR does not explicitly train to capture temporal patterns (*e.g.*, seasonal recurrences), it learns to align with the document distribution observed in corpora, which may naturally encode temporal patterns.

Specifically, we investigate document distribution across a monthly set from two TPOUR retrievers (January and June, 2023). The result of document distribution, computed as the ratio of retrieved to total documents per month, is in Tab. 18. We observe the January retriever favors winter months, while the June retriever favors summer months across years. This shows TPOUR's sensitivity to seasonal patterns without explicit supervision.

*Table 18.* Monthly document distribution of TPOUR-trained retrievers. We report monthly retrieval frequencies for two retrievers trained at different checkpoints (January 2023 and June 2023). The January retriever exhibits stronger alignment with winter months (*e.g.*, December–February), while the June retriever favors summer months (*e.g.*, May–August). This shows TPOUR can internalize seasonal patterns present in the training corpus without being explicitly trained for temporal recurrences.

| Year | Jan | Feb | Mar | Apr | May | Jun | Jul | Aug | Sep | Oct | Nov | Dec |
|---|---|---|---|---|---|---|---|---|---|---|---|---|
| | | | | | TPOUR Contriever (January, 2023) | | | | | | | |
| 2022 | 0.45 | 0.448 | 0.403 | 0.438 | 0.419 | 0.365 | 0.44 | 0.517 | 0.536 | 0.534 | 0.524 | 0.656 |
| 2023 | 0.628 | 0.551 | 0.475 | 0.43 | 0.466 | 0.511 | 0.452 | 0.405 | 0.438 | 0.464 | 0.463 | 0.513 |
| Total | 1.078 | 0.999 | 0.878 | 0.868 | 0.885 | 0.876 | 0.892 | 0.922 | 0.974 | 0.998 | 0.987 | 1.169 |
| | | | | | TPOUR Contriever (June, 2023) | | | | | | | |
| 2022 | 0.27 | 0.333 | 0.245 | 0.326 | 0.376 | 0.472 | 0.458 | 0.419 | 0.387 | 0.383 | 0.399 | 0.402 |
| 2023 | 0.439 | 0.373 | 0.424 | 0.394 | 0.459 | 0.559 | 0.563 | 0.368 | 0.431 | 0.505 | 0.478 | 0.449 |
| Total | 0.709 | 0.706 | 0.669 | 0.72 | 0.835 | 1.031 | 1.021 | 0.787 | 0.818 | 0.888 | 0.877 | 0.851 |

## E. Qualitative Case Studies

### E.1. Temporal Preference Learning Without Explicit Time Expressions

To illustrate how TPOUR captures temporal preferences without explicit timestamp expressions, we present a qualitative case study using the Wikipedia article *Office 1 Superstore*. This example shows how semantic changes across document versions serve as implicit temporal signals.

Tab. 19 compares three versions of the same document from the 2018, 2021, and 2023 Wikipedia dumps used in TPOUR 's training set. The 2018 version describes contraction following the 2008 economic crisis, including market exits and a shift to e-commerce. The 2021 version reflects a structural change, emphasizing the 2018 acquisition by Panda Cooperation. By 2023, the company is portrayed as having re-expanded globally under Panda's ownership.

Notably, none of these documents contain explicit temporal information such as year strings. The distinctions arise solely from semantic content. TPOUR 's preference-based training setup contrasts such temporally distinct documents, enabling the model to learn implicit temporal alignment cues. As shown in Tab. 8, fewer than 2.5% of training documents include explicit year references, underscoring the importance of implicit signals in learning temporal preferences.

*Table 19.* Three versions of the same document are used in TPOUR training. Although no explicit timestamp strings appear in the document content, the semantic update—*retrenchment* (2018), *ownership transfer* (2021), and *re-expansion* (2023)—shows real-world temporal progression. TPOUR leverages such a document update to learn temporal preference without explicit supervision.

| Timestamp | Training Document Example (Title: *Office 1 Superstore*) |
|---|---|
| 2018-09-16 | **Office 1 Superstores International Inc. (OFFICE 1)** was founded in 1994 as a franchise retail chain selling office products and supplies, including office furniture and electronics. The company is headquartered in West Palm Beach, Florida, with international operations run from a central office and warehouse in Sofia, Bulgaria. The company uses multiple channels of distribution to reach customers, including retail stores, telemarketing, direct mail, e-commerce, and contract sales. **OFFICE 1 expanded its operations through master franchises in Europe, Asia, Africa, Latin America, and the Caribbean, and at its peak had stores in 25 countries. Post the 2008 economic crisis**, the company retrenched and closed vulnerable markets such as Italy, Slovenia, and Iceland, **shifting focus to e-commerce.** It entered France (2010) and Germany (2011) through joint ventures. |
| 2021-11-06 | **Office 1 International Inc. (Office 1)** is an international franchise company established in Florida, USA, and present in three countries—Bulgaria, France, and Greece. **On February 20, 2018, Panda Cooperation officially acquired all trademark rights** of the Office 1 Superstore portfolio. From a major franchisee in Bulgaria, **Panda Cooperation became the sole owner and representative of Office 1 brands worldwide**. In 1998, Panda had received a master franchise for Bulgaria, and by 2021, **Office 1 Superstore was the largest office supply chain in Bulgaria, serving over 130,000 business clients**. |
| 2023-11-06 | **Office 1 International Inc. (Office 1)** is an international franchise company established in Florida, USA, and currently present in **27 countries** including Bulgaria, France, and Greece, with over **600 locations**. **Office 1 was founded in 1989 by Mark Baccash**. Panda Cooperation, having acquired all Office 1 trademark rights in 2018, remains the **sole global owner and operator**. Office 1 maintains an extensive store network in Bulgaria and has expanded its online presence through multiple social media accounts. |

## E.2. Comparative Analysis of Retrieved Documents

*Table 20.* Retrieved documents comparison between TPOUR Contriever (2021) and Contriever for three example queries. The text containing the correct answers is highlighted in **bold**.

| Model | Rank | Document | Timestamp |
|---|---|---|---|
| **Query:** Who has won the most Olympic medals in curling as of 2021? | | | |
| TPOUR Contriever | Top 1 | Brad Gushue // [...] Defeating **Edin** in the final. [...] Defeating Scotland's Bruce Mouat in the final. [...] | 2021-11-30 |
| | Top 2 | United States Curling Association // [...] Skip **John Shuster**'s team won the gold medal. **John Shuster** [...] | 2021-11-30 |
| Contriever | Top 1 | Canada at the Olympics // [...] Jones, Kaitlyn **Lawes**, Jill Officer, Dawn McEwen and spare Kirsten Wall went unbeaten [...] | 2018-12-06 |
| | Top 2 | Canada at the Olympics // [...] Jones, Kaitlyn **Lawes**, Jill Officer, Dawn McEwen and spare Kirsten Wall went unbeaten [...] | 2020-11-27 |
| **Query:** Who is the No. 1 ranked tennis player in the world as of 2021? | | | |
| TPOUR Contriever | Top 1 | Juan Martin del Potro // [...] Lost his quarterfinal against world number 1 **Novak Djokovic** [...] | 2021-12-11 |
| | Top 2 | Tennis in Spain // [...] Tying him with Federer and **Novak Djokovic**. [...] | 2021-11-09 |
| Contriever | Top 1 | Tennis // [...] **Novak Djokovic**, a rival of both Nadal and Federer, is also [...] | 2020-12-06 |
| | Top 2 | Alexander Zverev // [...] **Novak Djokovic** has said, "Hopefully, he can surpass me." [...] | 2018-12-12 |
| **Query:** What is the current macOS operating system as of 2021? | | | |
| TPOUR Contriever | Top 1 | macOS // [...] **macOS Monterey was presented as version 12 in 2021**. [...] | 2021-12-05 |
| | Top 2 | macOS Server // [...] macOS 12 (Server 5.12) [...] Operates on **macOS Monterey (12)** and later. [...] | 2021-12-15 |
| Contriever | Top 1 | Personal Computer // [...] macOS is a Unix-based graphical operating system, and [...] | 2018-12-15 |
| | Top 2 | macOS // [...] **macOS Monterey was presented as version 12 in 2021**. [...] | 2021-12-05 |

*Table 21.* Retrieved documents comparison between TPOUR Contriever (2018) and Contriever for three example queries. The text containing the correct answers is highlighted in **bold**.

| Model | Rank | Document | Timestamp |
|---|---|---|---|
| **Query:** When did the Golden State Warriors win the Finals as of 2018 | | | |
| TPOUR Contriever | Top 1 | Willie Green // [...] defeated the Cleveland Cavaliers in four games of the **2018 NBA Finals**. [...] | 2018-11-25 |
| | Top 2 | Jarron Collins // [...] Collins won his third championship in four years when the Warriors defeated the Cleveland Cavaliers in the **2018 NBA Finals**. [...] | 2019-12-27 |
| Contriever | Top 1 | National Basketball Association Criticisms and Controversies // [...] Some NBA fans have accused the league of conspiring to have large-market teams [...] | 2019-12-30 |
| | Top 2 | NBA Finals // [...] The Warriors swept the Cavaliers 4-0 [...] | 2020-12-11 |
| **Query:** What NFL player has the most NFL rings as of 2018 | | | |
| TPOUR Contriever | Top 1 | NFL Top 100 Players of 2018 // [...] It ended with reigning NFL MVP **Tom Brady** being ranked #1 [...] | 2018-12-07 |
| | Top 2 | Jeff Stoutland // [...] Stoutland won his first Super Bowl ring when the Eagles defeated the New England Patriots in Super Bowl LII. [...] | 2020-12-19 |
| Contriever | Top 1 | Super Bowl Ring // [...] The New England Patriots' Super Bowl XLIX rings reportedly cost $36,500 each [...] | 2019-12-30 |
| | Top 2 | Super Bowl Ring // [...] Super Bowl LI ring has 283 diamonds, to commemorate their comeback [...] | 2020-12-19 |
| **Query:** When did the Philadelphia Eagles play in the Super Bowl last as of February 23, 2018 | | | |
| TPOUR Contriever | Top 1 | Curse of Billy Penn // [...] On February 4, 2018, the Philadelphia Eagles defeated the New England Patriots in **Super Bowl LII** 41-33 [...] | 2018-12-07 |
| | Top 2 | Jeff Stoutland // [...] Stoutland won his first Super Bowl ring when the Eagles defeated the New England Patriots in **Super Bowl LII**. [...] | 2020-12-19 |
| Contriever | Top 1 | 2018 Philadelphia Eagles Season // [...] A new Super Bowl champion would be crowned. [...] | 2020-12-19 |
| | Top 2 | Sports-Related Curses // [...] The Eagles accumulated a lot of playoff heartbreak, including 2 Super Bowl losses [...] | 2020-12-19 |

# F. Discussion on Future Work

## F.1. Relaxation of Temporally Distributed Corpora

As noted in Sec. 5, TPOUR requires temporally distributed corpora (*e.g.*, Wikipedia dumps). Each dump is treated as a snapshot of world knowledge at a specific point in time (Jatowt et al., 2005). While documents may mention events from various eras, their dominant temporal context aligns with the collection period (*e.g.*, the phrase "last week" in a 2020 dump naturally grounds to that year). This assumption allows TPOUR to induce temporal preferences at the corpus level without requiring document-level timestamp supervision.

Such versioned corpora may not always be available in practice. However, we believe that utilizing coarse-grained temporal signals is a promising future direction. Coarse-grained temporal signals often exist in other domains. For example, user-generated content typically carries internal timestamps (*e.g.*, server logs or metadata), even if not explicitly exposed.

The central insight of TPOUR is that even minimal corpus-level temporal signals can be sufficient to induce temporal awareness in retrievers, without relying on explicit document-level timestamps. Moreover, document-level annotations, while useful, are often noisy, missing, or inconsistent due to edits, revisions, or formatting errors (Dhingra et al., 2022).

## F.2. Analysis of Temporal Grounding

In practice, temporal grounding is expected to occur at the time of querying (or inference), reflecting the user's current context for implicit queries. We first conducted a preliminary experiment to test whether a TPOUR-trained retriever optimized to predict more recent times can surpass general retriever baselines (*e.g.*, Contriever, Nomic Embed v2 MoE). To empirically validate this assumption, we evaluated the TPOUR-trained Contriever (2021) on RealtimeQA (2023) by aggregating all

monthly test sets from RealtimeQA. Tab. 22 shows that the TPOUR-trained Contriever (2021) outperforms general retrievers (*e.g.*, Contriever and Nomic Embed v2 MoE) when the test set contains 2023-related queries. This shows that TPOUR can train retrievers to handle recent queries better than general-purpose retrievers. To further analyze the impact of temporal grounding, we categorized RealTimeQA queries along two different axes. We used GPT-4o (OpenAI et al., 2024) to assign each of the 1,428 queries to both a (1) Temporal Category and a (2) Topic Category. We then manually reviewed all queries to ensure accurate classification. Queries from underrepresented topic categories (fewer than 30 examples) were grouped under "Others" to stabilize analysis. Detailed information on each category is shown at Tab. 23 and Tab. 24.

Given these queries assigned to each temporal/topic category, we evaluated NDCG@5 (N@5) performance across categories. Here, $\Delta$ represents the score difference between TPOUR Contriever (2021) and baseline Contriever. Tab. 25 and 26. The temporal category results show an interesting insight. TPOUR Contriever (2021) is especially effective on "Timeless" temporal queries, with smaller improvements for "Distant Past" queries. In terms of topic category, timely categories such as "Sports" and "Business" benefited the most, while "Health" and "Environment" showed relatively smaller performance gains over Contriever. Given the per-query $\Delta$, we further investigate a case study to examine which examples TPOUR Contriever (2021) performs better on compared to Contriever in Tab. 27. It shows that TPOUR Contriever (2021) outperforms Contriever on queries requiring temporal grounding by retrieving contextually and temporally aligned documents.

*Table 22.* Performance of the TPOUR-trained retriever aligned to recent time (TPOUR Contriever (2021)), which surpasses general retrievers (*e.g.*, Contriever and Nomic Embed v2 MoE).

| RealtimeQA (2023) | N@5 | N@10 |
|---|---|---|
| Contriever | 44.39 | 45.25 |
| Nomic Embed v2 MoE | 35.20 | 35.88 |
| TPOUR Contriever (2018) | 22.48 | 23.92 |
| TPOUR Contriever (2021) | **48.43** | **51.22** |

*Table 23.* Topic categories. RealTimeQA (2023) queries are categorized into topical domains such as Sports, Business and Health. Queries from underrepresented domains are grouped under Others.

| Category | # Queries |
|---|---|
| Sports | 122 |
| Business | 119 |
| International | 224 |
| Entertainment | 114 |
| Politics | 217 |
| Environment | 61 |
| Health | 99 |
| Others | 472 |

*Table 24.* Temporal categories. RealTimeQA (2023) queries are categorized as Timeless, Recent Past, Immediate, or Distant Past based on their temporal information. Most queries fall into the "Timeless" category, which requires retrieving temporally up-to-date documents.

| Category | # Queries | Description / Example |
|---|---|---|
| Timeless | 687 | The query does not mention time, but requires up-to-date documents. *e.g.*, "Which Covid-19 variant of Omicron become the most dominant in US?" |
| Recent Past ($\leq$ 1 year) | 165 | Explicitly references events from the recent past (*e.g.*, "last year"). *e.g.*, "How many flights on private jets were made globally last year?" |
| Immediate | 504 | Refers to ongoing or very recent events (*e.g.*, "this week"). *e.g.*, "The U.S. embassy in which country was evacuated this week?" |
| Distant Past ($>$ 1 year) | 72 | Refers to events that occurred more than a year ago (*e.g.*, "after the 2020"). *e.g.*, "Dominion Voting Systems settled with which TV network in a defamation lawsuit over the broadcast of lies after the 2020 presidential election?" |

*Table 25.* Temporal category performance. TPOUR Contriever (2021) is effective on "Timeless" (+7.36) compared to baseline Contriever, while showing smaller gains on "Distant Past" (+2.23).

| Model | Timeless | Recent Past ($\leq$1 year) | Immediate | Distant Past (>1 year) |
|---|---|---|---|---|
| Contriever | 46.08 | 44.50 | 43.14 | 50.81 |
| Nomic Embed v2 MoE | 37.50 | 35.16 | 32.69 | 40.04 |
| TPOUR Contriever (2018) | 23.17 | 23.05 | 21.30 | 23.72 |
| TPOUR Contriever (2021) | 53.44 | 50.53 | 48.05 | 53.04 |
| $\Delta$ over Contriever | +7.36 | +6.03 | +4.91 | +2.23 |

*Table 26.* Topic category performance. Timely categories such as "Sports" (+10.81) and "Business" (+7.39) benefited the most from TPOUR Contriever (2021), while "Health" (+3.06) and "Environment" (+4.71) showed relatively smaller gains over baseline Contriever.

| Model | Sports | Business | International | Entertainment | Politics | Environment | Health | Others |
|---|---|---|---|---|---|---|---|---|
| Contriever | 42.64 | 41.98 | 45.07 | 45.18 | 45.29 | 44.64 | 51.11 | 46.07 |
| Nomic Embed v2 MoE | 34.45 | 33.36 | 36.76 | 33.72 | 37.96 | 35.78 | 35.29 | 37.28 |
| TPOUR Contriever (2018) | 19.16 | 24.37 | 20.79 | 20.31 | 21.92 | 25.39 | 29.81 | 23.32 |
| TPOUR Contriever (2021) | 53.45 | 49.37 | 51.79 | 51.26 | 50.48 | 49.35 | 54.17 | 51.88 |
| $\Delta$ over Contriever | +10.81 | +7.39 | +6.72 | +6.08 | +5.19 | +4.71 | +3.06 | +5.81 |

*Table 27.* Example queries across temporal and topic categories. Each example illustrates how TPOUR Contriever (2021) outperforms baseline Contriever, with improvements ranging from +48.52 (Environment, Recent Past) to +86.88 (Entertainment, Immediate).

| Query | Temporal Category | Topic Category | $\Delta$ over Contriever |
|---|---|---|---|
| The Biden administration is monitoring a potentially major labor strike brewing in which industry? | Timeless | Politics | +69.92 |
| Newly released figures show that the amount of electricity produced by which type of renewable energy hit a record high in Britain last year? | Recent Past | Environment | +48.52 |
| The nominees for the 75th Emmy Awards television's top honor were announced this week. Which show received the most nominations? | Immediate | Entertainment | +86.88 |
| China's birth rate declined for the first time in decades in 2022. It has been the world's most populous nation since at least when? | Distant Past | International | +78.60 |

## F.3. Appropriate $\alpha$ Selection

Determining the optimal interpolation weight $\alpha$ is a non-trivial problem. We assume temporal grounding for each query, determined by either explicit or implicit temporal intent. This offers an advantage over using a single "global" retriever to handle queries from multiple time periods. **Reduced training burden.** Avoids forcing a single model to learn both semantic and temporal alignment simultaneously. **Temporal sensitivity.** A global retriever must balance signals across many time periods, which can weaken or distort its sensitivity for specific periods. **Modularity.** We can decouple the problem into two subproblems. (1) Router to detect a query's temporal intent and (2) Retriever to retrieve temporally aligned documents. **Interpretability.** Interpolation weights $\alpha$ make it easy to trace how retrieval preferences shift across time.

**For explicit temporal queries** (*e.g.*, "in 2019"), tools like dateparser (Scrapinghub) can be used to extract the timestamp, which directly maps $\alpha$ to select or interpolate among TPOUR retrievers. **For implicit temporal queries**, we distinguish two types: (1) Queries referring to the current time (*e.g.*, "Who is the current prime minister?", "What time is it?"). In such cases, defaulting to the most recent TPOUR retriever is a viable approach, under the assumption that users intend to refer to the present. TPOUR Contriever (2021), despite being trained two years earlier, still outperforms general-purpose retrievers on the RealTimeQA (2023) benchmark, as shown in Tab. 22. (2) Queries implying a specific but unstated time (*e.g.*, "When was the 21st conference held?"). In these cases, training and using a query intent classifier to predict the optimal $\alpha$ is feasible. (Wu et al., 2024) has already demonstrated that predicting query timestamps is possible, achieving 96% test accuracy.

# G. Notations

*Table 28.* Definitions of notations used in the above formalizations.

| Symbol | Definition |
|---|---|
| $Q$ | Query text |
| $D$ | Document text |
| $D^+$ | Positive document |
| $D^-$ | Negative document |
| $D^t$ | Temporally aligned document |
| $D^{t'}$ | Temporally misaligned document |
| $S(\cdot, \cdot)$ | Similarity function |
| $S_\theta(y^w)$ | Abbreviated form of $S(\pi_\theta(Q), \pi_\theta(D^t))$ |
| $S_\theta(y^l)$ | Abbreviated form of $S(\pi_\theta(Q), \pi_\theta(D^{t'}))$ |
| $\pi_q$ | Query encoder |
| $\pi_k$ | Document encoder |
| $\pi_{\text{ref}}$ | Reference policy (encoder) |
| $\pi_\theta$ | Training target policy (encoder) |
| $\pi_\theta^t$ | Training target policy (encoder) that aligned at time $t$ |
| $\mathcal{L}(\cdot)$ | Loss function |
| $\mathcal{L}_{\text{TRPO}}(\cdot)$ | TRPO loss |
| $\mathcal{L}_{\text{CE}}(\cdot)$ | Contrastive loss |
| $\mathcal{L}_{\text{total}}(\cdot)$ | Total loss |
| $m$ | Momentum hyperparameter |
| $\lambda$ | $\mathcal{L}_{\text{TRPO}}$ and $\mathcal{L}_{\text{CE}}$ balance hyperparameter |
| $\alpha$ | Time vector interpolation hyperparameter |
| $\theta_q$ | Query (policy) encoder weight |
| $\theta_k$ | Document (policy) encoder weight |
| $\theta_{\text{ref}}$ | Reference policy weight |
| $\theta$ | Training target policy ($\pi_\theta$) weight |
| $\theta_{\text{base}}$ | Base pretrained encoder weight |
| $\theta_t$ | The encoder weight fine-tuned on data from time period $t$ |
| $y^w$ | Preferred output |
| $y^l$ | Less preferred output |
| $x$ | Prompt input |
| $\sigma(\cdot)$ | Sigmoid function |
| $\beta$ | DPO temperature parameter |
| $T$ | Contrastive loss temperature parameter |
| $\tau_t$ | Time vector for time $t$ |
| $t_{start}$ | Start time period |
| $t_{mid}$ | Middle time period |
| $t_{end}$ | End time period |

