# OpenReview forum: "Temporal Preference Optimization for Unsupervised Retrieval"
_ICML.cc/2026/Conference — ICML 2026 regular_

### Official Review · Reviewer_L6CF · 2026-03-11

**Soundness:** 2
**Presentation:** 3
**Significance:** 2
**Originality:** 3
**Overall Recommendation:** 4
**Confidence:** 3

**Summary:**

This paper identifies a temporal misalignment issue in unsupervised dense retrievers when applied to mixed-timestamp document collections: retrievers may return semantically relevant but temporally inconsistent documents. To address this problem, the paper proposes TPOUR, an unsupervised training framework that incorporates temporal preference into contrastive learning via Temporal Retrieval Preference Optimization (TRPO). The method encourages the retriever to favor temporally aligned documents while preserving semantic similarity. Experiments on temporal QA tasks show that TPOUR outperforms existing baselines and that the learned temporal embeddings allow continuous generalization to unseen intermediate time periods through interpolation.

**Compliance With Llm Reviewing Policy:**

Affirmed.

**Final Justification:**

The author's two rebuttals effectively alleviated most of the weaknesses I previously mentioned, so I ultimately adjusted the score to 4.

**Key Questions For Authors:**

1. Comparison with instruction-aware retrieval.
If LLM-based and instruction-tuned dense retrievers can enforce temporal alignment through simple instructions, does the proposed training framework still provide advantages?
2. Motivation for TRPO.
Given that temporal alignment is a deterministic binary signal, why is TRPO preferable to adding a temporal contrastive objective on top of standard contrastive learning?
3. Validity of the interpolation assumption.
How do the authors verify that the training data satisfies the “distributional similarity” assumption required for the proposed time-embedding interpolation method?

**Limitations:**

yes

**Strengths And Weaknesses:**

Strengths
1.  Meaningful problem. The paper highlights the temporal misalignment issue in current retrieval models, which is a practical and relevant problem.
2. Clear presentation. The paper is well written and easy to follow, with a clear progression from problem formulation to experiments.
3. Informative visualizations. The figures (e.g., architecture diagrams and interpolation heatmaps) effectively illustrate the proposed ideas.

Weaknesses
1. Missing comparisons with instruction-aware retrievers. The experiments do not include recent LLM-based and instruction-tuned retrieval models [1-2], which may be able to control temporal alignment through simple instructions.
2. Unclear motivation for TRPO. Temporal alignment is a deterministic binary signal rather than a subjective preference, making the use of a preference optimization framework [3] insufficiently motivated.
3. Insufficient ablation for TRPO. The paper does not provide enough ablation experiments to verify whether TRPO is necessary compared to simpler alternatives such as a temporal contrastive loss.
4. Weak theoretical support for temporal interpolation. The interpolation method relies on the “distributional similarity” assumption in Appendix B.2, but the paper does not verify whether this assumption holds in the training data.


References

[1] FollowIR: Evaluating and Teaching Information Retrieval Models to Follow Instructions. NAACL 2025.

[2] Towards Better Instruction Following Retrieval Models. arXiv 2025.

[3] Direct Preference Optimization: Your Language Model is Secretly a Reward Model. NeurIPS 2023.

---

> ### Author Rebuttal · Authors · 2026-03-30
>
> We thank the reviewer for the insightful comment and we have provided our responses below.
>
> ---
> >W1/Q1. Comparison with instruction-aware retrieval.
>
> - We clarify that our results show that **instruction-aware retrieval (Qwen3-Embedding-8B [1]) provides partial gains, but does not fully mitigate temporal misalignment, even with a model ~72.7× larger than TPOUR**.
>
> - We use the following prompt to incorporate temporal constraints.
>
> ```
> [trimmed due to space limit]
> Instructions:
> 1. Identify the temporal intent of the query.
> 2. Filter or downweight documents that violate the temporal constraint.
> 3. Rank documents by both semantic relevance and temporal alignment.
> 4. Prefer documents whose timestamps are closest to (but not exceeding) the target time.
>
> Query: {QUERY}
> ```
> The results show that instruction-aware prompting improves performance over the naive baseline (e.g., 2018 N@5: 29.57 to 32.69). But, it still underperforms TPOUR (44.10), which achieves substantially higher performance. **This leads to two key observations:**
>
> **(1) Instruction-aware retrieval provides partial gains but is insufficient for temporal alignment.**
> - While instructions help guide retrieval, they do not fully resolve temporal grounding when the underlying embeddings are not temporally aligned.
>
> **(2) TPOUR and instruction-aware retrieval are complementary rather than substitutive.**
> - Instruction-aware methods rely on instructions.
> - By contrast, TPOUR enables the retriever to learn temporal alignment directly from the corpus, allowing it to handle both explicit and implicit temporal queries.
>
> Table. SituatedQA Explicit Result.
> |Explicit|2018 N@5|2018 N@10|2021 N@5|2021 N@10|
> |---|---|---|---|---|
> |Qwen3-Embedding-8B|29.57|33.12|34.32|37.54|
> |Qwen3-Embedding-8B (Instruction-Aware)|32.69|35.48|38.51|40.64|
> |TPOUR|**43.93**|**46.66**|**40.21**|**44.72**|
>
> - We will include additional comparisons in the revision.
> ---
> >W2/Q2 Why does TRPO use preference optimization for deterministic binary signal?
>
> Temporal alignment is not a subjective preference. However, following prior work [2], **we treat temporal relevance as an implicit pairwise preference signal and this signal is derived from corpus-level temporal differences**.
>
> This provides two advantages.
>
> **(1) It directly models relative temporal document ordering (i.e., ranking)**.
> - TRPO optimizes pairwise preferences, explicitly encouraging correct temporal ranking.
>
> **(2) It is more robust to noisy and implicit supervision**.
> - Temporal signals are often indirect from corpus level documents and noisy. Pairwise preferences enable consistent learning without requiring clean binary labels.
>
> We will clarify this motivation more explicitly in the revision.
>
> ---
> >W3. TRPO ablation with simpler alternatives such as a temporal contrastive loss.
>
> Thank you for the valuable suggestion on comparing TRPO with simpler alternatives. However, we would like to clarify the following.
>
> **(1) Temporal contrastive baselines in prior work.**
> - To the best of our knowledge, there is no widely adopted temporal contrastive loss designed for dense retrieval. The related approach is **time-aware contrastive retrieval (e.g., TimeR4), which we already include as a baseline**.
>
> **(2) We actually provided an ablation isolating TRPO vs. contrastive learning.**
> - In Fig.10, we interpolate between the contrastive loss and the TRPO ($\lambda = 1.0$ contrastive-only, $\lambda = 0$ TRPO-only) and show performance consistently peaks at moderate $\lambda$.
>
> These results show that **TRPO contributes beyond contrastive learning** which leads to the best performance.
>
> We will clarify this ablation and its implications more explicitly in the revision.
>
> ---
> >W4/Q3. How do the authors verify that the training data satisfies the "distributional similarity" assumption?
>
> Thank you for raising this important point. We would like to clarify the following.
>
> **(1) The interpolation assumption is grounded in prior work**.
>
> This assumption is not introduced from scratch, but **builds on findings from the time vector literature** [3]. Our work extends this idea to encoder-based retrieval.
>
> **(2) We provide empirical evidence from our dataset**.
>
> As shown in Appendix Tab.5, **a large portion of documents with the same topic are shared across timestamps** (Filtered Intersection), while only a subset consists of entirely new content (Unique), accounting for less than 30% of the total documents.
>
> We will clarify this assumption and supporting evidence more explicitly in the revision.
>
> ---
> We hope our response addresses your questions and appreciate your acknowledgment of our problem setup. Please let us know if you have any remaining questions.
>
> [1] Qwen3 Embedding: Advancing Text Embedding and Reranking Through Foundation Models, arXiv, 2025
>
> [2] Direct Preference Optimization: Your Language Model is Secretly a Reward Model, NeurIPS, 2023
>
> [3] Time is Encoded in the Weights of Finetuned Language Models, ACL, 2024

---

> > ### Author Rebuttal · Reviewer_L6CF · 2026-04-03
> >
> > Thank you for the authors' detailed response, which has addressed most of my concerns fairly well. However, I still have reservations regarding W3, and my questions are as follows.
> >
> > According to Line 210 in the paper, the training in this work uses conventional CE loss + TRPO loss. The point I am questioning is whether simply replacing the TRPO loss here with a temporal contrastive loss would yield better performance than the TRPO loss.
> >
> > When implementing TRPO, the authors construct a batch of positive and negative samples. I believe this batch of data could be directly used for temporal contrastive loss — that is, the training scheme I have in mind is CE loss + Temporal CE loss.
> >
> > (1) I am not certain whether the "Temporal contrastive baselines in prior work" mentioned by the authors refers to this specific setting.
> >
> > (2) The ablation experiment conducted by the authors in Fig. 10 only retains CE loss, which merely demonstrates that performance is poor when only CE loss is used.
> >
> > Nevertheless, the authors' detailed response has resolved most of my concerns. I now consider this paper to be of borderline quality. If the authors can provide further clarification regarding W3 through experiments or other reasonable means, I would be very willing to raise my score.

---

> > > ### Author Response · Authors · 2026-04-07
> > >
> > > Thank you for this thoughtful and clarifing the suggestion regarding temporal contrastive baseline. We appreciate the reviewer's suggestion to compare against a temporal contrastive baseline (i.e., CE loss + Temporal CE loss), which we agree is an important and relevant.
> > >
> > > ---
> > > **Temporal Contrastive Baseline**
> > >
> > > **(1) Setup**
> > > - Following the reviewer's suggestion, we implement a temporal contrastive baseline. For each query, we construct positive documents $d^+$ (temporally aligned) and negative documents $d^-$ (temporally misaligned). We optimize a contrastive objective (Temporal CE loss) that encourages higher similarity between the query and $d^+$ than $d^-$.
> > >
> > > - Thus, the final objective is $L_{\text{total}} = \lambda L_{\text{CE}} + (1 - \lambda)L_{\text{TemporalCE}}$. We tune $\lambda$ over a grid search (0.0 to 1.0) and report the best performance (i.e., $\lambda=0.5$) for a fair comparison.
> > >
> > > **(2) Result**
> > > - As shown in Tables 1 and 3, TPOUR consistently outperforms the temporal contrastive baseline on SituatedQA (2018, 2021), achieving stronger performance across all settings.
> > > - It also retrieves a substantially higher proportion of temporally aligned documents (Tables 2 and 4).
> > >
> > > Table 1. SituatedQA explicit result on TRPO vs. Temporal Contrastive baseline.
> > > |SituatedQA (Explicit)|2018 N@5|2018 N@10|2021 N@5|2021 N@10|
> > > |-|-|-|-|-|
> > > |Contriever|29.30|33.35|37.85|41.05|
> > > |Temporal Contrastive|35.00|38.60|37.32|42.11|
> > > |TRPO|**43.93**|**46.66**|**40.21**|**44.72**|
> > >
> > > Table 2. SituatedQA explicit normalized proportion (%) of retrieved documents that match the target year (2018, 2021), conditioned on the test set year.
> > > |Retrieved / Total (%)|2018|2021|
> > > |-|-|-|
> > > |Temporal Contrastive|26.1|37.9|
> > > |TRPO|**37.3**|**56.2**|
> > >
> > >
> > > Table 3. SituatedQA implicit Result on TRPO vs. Temporal Contrastive baseline.
> > > |SituatedQA (Implicit)|2018 N@5|2018 N@10|2021 N@5|2021 N@10|
> > > |-|-|-|-|-|
> > > |Contriever|29.89|34.60|33.06|37.08|
> > > |Temporal Contrastive|36.55|39.38|33.70|38.08|
> > > |TRPO|**44.11**|**46.59**|**39.40**|**44.72**|
> > >
> > > Table 4. SituatedQA implicit normalized proportion (%) of retrieved documents that match the target year (2018, 2021), conditioned on the test set year.
> > > |Retrieved / Total (%)|2018|2021|
> > > |-|-|-|
> > > |Temporal Contrastive|26.7|31.0|
> > > |TRPO|**34.9**|**37.7**|
> > >
> > > We believe this performance difference arises from how temporal signals are utilized. Prior work has shown that preference-based ranking objectives are effective for learning from implicit, relative supervision, especially when absolute labels are difficult to define or noisy [1,2]. In our setting, temporal relevance is often implicit and noisy due to corpus level signal, making strict positive/negative contrastive supervision may harder to optimize.
> > >
> > > By contrast, TRPO uses preference signals over temporally aligned versus misaligned documents, which we hypothesize provides a more flexible learning signal for capturing such implicit temporal structure.
> > >
> > > We will include this additional experiment and clarification in the final version.
> > >
> > > [1] Query Chains: Learning to Rank from Implicit Feedback
> > >
> > > [2] Reinforcement Learning from Human Feedback: A Statistical Perspective

---

### Official Review · Reviewer_7vgB · 2026-03-12

**Soundness:** 3
**Presentation:** 3
**Significance:** 2
**Originality:** 3
**Overall Recommendation:** 4
**Confidence:** 3

**Summary:**

This paper addresses temporal misalignment in retrieval with TPOUR, which adds a preference-based temporal loss (TRPO) to unsupervised contrastive retrieval training and then uses time-vector interpolation to generalize across periods. The problem is real and the reported gains on the custom temporal QA benchmarks are meaningful, but the overall contribution feels more specialized.

**Compliance With Llm Reviewing Policy:**

Affirmed.

**Final Justification:**

All my concerns have been addressed.

**Key Questions For Authors:**

- Do recent large embedding models (such as Qwen3-Embedding-8B or other strong embedding models) already mitigate temporal misalignment?

**Limitations:**

yes

**Strengths And Weaknesses:**

Strengths:
- The paper focuses on a real retrieval failure mode. The motivation that time-unaware retrievers can retrieve semantically related but temporally wrong documents is clear, and the examples in the introduction are easy to follow.
- The empirical gains on SituatedQA and RealTimeQA are non-trivial. Main results show clear improvements over Contriever and other baselines on both explicit and implicit temporal queries, especially when the retrieval model is aligned to the relevant period.
- The paper goes beyond a single result table by looking at timestamp prediction and interpolation behavior, which helps show that the model is learning some temporally structured representation rather than only overfitting the benchmark.

Weakness:
- On the modeling side, the contribution appears limited because TPOUR mainly combines existing contrastive retrieval with a preference-style objective over temporally constructed pairs, so the novelty seems closer to a specialized data construction and training setup than to a substantial new ML method. The paper would be stronger if it more clearly isolated what TRPO contributes beyond standard pairwise ranking or supervised contrastive learning using the same aligned/misaligned pairs.
- The baseline comparison is also incomplete for the claimed problem setting, since practical alternatives such as query rewriting, document rewriting, or metadata-aware reranking are not evaluated. Adding these baselines would clarify whether the gains come from a better retrieval objective or simply from introducing temporal grounding that simpler pipeline methods may already provide.

---

> ### Author Rebuttal · Authors · 2026-03-30
>
> Thank you for the thoughtful feedback. We provide our responses below.
>
> ---
> > W1. What TRPO contributes beyond standard pairwise ranking?
>
> We clarify that **our contribution is an embedding-level preference learning objective for temporal alignment in unsupervised retrieval**.
>
> **(1) To the best of our knowledge, this is one of the first efforts to temporally align retrievers at the embedding level.**
> - Standard unsupervised contrastive retrieval optimizes solely for semantic similarity while TRPO injects temporal preference signals directly into the retriever's embedding space using aligned and misaligned document pairs.
>
> **(2) Our contribution goes beyond temporal pair construction.**
> - Our approach enables training an unsupervised retriever with temporal preference signals without requiring explicit fine-grained labels for every document.
>
> **(3) We respectfully note that we already compared semantic contrastive loss alone and with TRPO, which helps improve performance.**
> - Fig.10 varies the interpolation between $𝓛_{\text{TRPO}}$ ($λ = 0$) and $𝓛_{\text{CE}}$ ($λ = 1$).
>
> Performance peaks at moderate $\lambda$, not at $\lambda = 1.0$ (contrastive-only). Gains reach +21.60 (2018) and +4.17 (2021) nDCG@5 over contrastive-only.
>
> We value your perspective and will revise our paper to more explicitly discuss our contribution and what TRPO contributes beyond standard methods.
>
> ---
> >W2. The baseline comparison with alternatives (query rewriting, document rewriting, or metadata-aware reranking).
>
> Thank you for your thoughtful feedback on incorporating additional comparisons such as query rewriting. We would first like to clarify that **our method is orthogonal to these approaches and can be used in conjunction with them**.
>
> **(1) Metadata-aware reranking**.
> - Our method does not rely on any document-level metadata. Instead, **we rely solely on corpora collected at specific time periods, without requiring explicit timestamp annotations for each document**.
>
> **(2) Document rewriting**.
> - Document rewriting requires rewriting the entire corpus (e.g., full Wikipedia), **which introduces substantial computational overhead** [3].
>
> **(3) Query rewriting** [1].
> - We incorporate query rewriting into our method to evaluate its effectiveness. Specifically, we use GPT-OSS-20B [2] as a frozen rewriter, following prior prompting strategies.
> - **We also compare against Qwen3-Embedding-8B [4], which has significantly more parameters (~72.7× larger) than our model**.
>
> - An example of rewritten queries below:
> ```
> [Original]
> - who is the secretary of state for northern ireland as of January 09, 2021
> - where did the world's largest recorded wave occur as of 2018
>
> [Rewritten]
> - Secretary of State for Northern Ireland Jan 09 2021
> - world's largest recorded wave location 2018
> ```
> - The results show that **(1) Query rewriting does not consistently improve performance for either TPOUR or Qwen3-Embedding-8B. (2) TPOUR consistently outperforms the larger baseline (Qwen3-Embedding-8B), regardless of whether query rewriting is applied**. We report averages due to rebuttal space limit (~5000 char).
>
> Table. Query Rewriting (QR) on SituatedQA.
> |Avg (Explicit & Implicit)|2018 N@5|2018 N@10|2021 N@5|2021 N@10|
> |---|---|---|---|---|
> |Qwen3-Embedding-8B (QR)|35.02|38.46|33.12|35.86|
> |Qwen3-Embedding-8B|29.47|32.58|33.54|36.54|
> |TPOUR (QR)|40.16|43.31|37.68|38.80|
> |TPOUR|**44.02**|**46.63**|**39.81**|**44.73**|
>
> Table. Query Rewriting (QR) on RealtimeQA.
> |Avg (Explicit & Implicit)|Jan N@5|Jan N@10|Dec N@5|Dec N@10|
> |---|---|---|---|---|
> |Qwen3-Embedding-8B (QR)|22.97|23.72|40.22|37.79|
> |Qwen3-Embedding-8B|23.35|23.83|41.29|38.69|
> |TPOUR (QR)|29.61|28.93|45.50|43.32|
> |TPOUR|**31.90**|**31.52**|**48.77**|**45.88**|
>
> ---
> >Q1. Do recent large embedding models (e.g., Qwen3-Embedding-8B) mitigate temporal misalignment?
>
> We agree that recent embedding models have significantly improved general retrieval performance.
> **However, our empirical results show that temporal misalignment remains a significant challenge even for these models**.
>
> In our experiments (see W2), we directly compare against a strong large-scale embedding model, **Qwen3-Embedding-8B (~72.7× larger than ours)** and show that **TPOUR outperforms Qwen3-Embedding-8B across temporal retrieval benchmarks**.
>
> We will clarify this comparison and its implications more explicitly in the revised paper.
>
> ---
> We hope our response addresses your questions. We also appreciate your acknowledgment of the practical relevance of our problem setting and the strength of our empirical results. Please let us know if you have any remaining questions.
>
> [1] Query Rewriting in Retrieval-Augmented Large Language Models, EMNLP, 2023
>
> [2] gpt-oss-120b & gpt-oss-20b Model Card, arXiv, 2025
>
> [3] Cocktail: A Comprehensive Information Retrieval Benchmark with LLM-Generated Documents Integration, ACL-Findings, 2024
>
> [4] Qwen3 Embedding: Advancing Text Embedding and Reranking Through Foundation Models, arXiv, 2025

---

> > ### Author Rebuttal · Reviewer_7vgB · 2026-04-04
> >
> > I thank the authors for answering my questions. All my concerns have been addressed.

---

> > > ### Author Response · Authors · 2026-04-07
> > >
> > > We sincerely thank the reviewer for the constructive feedback. We are pleased that our responses addressed your concerns and update our score. We greatly appreciate your support and will incorporate the suggested improvements in the final version.

---

### Official Review · Reviewer_qX27 · 2026-03-17

**Soundness:** 3
**Presentation:** 2
**Significance:** 2
**Originality:** 2
**Overall Recommendation:** 4
**Confidence:** 2

**Summary:**

This work proposes TPOUR, a framework designed to tackle the temporal mismatch problem inherent in unsupervised dense retrievers. By introducing Temporal Preference Optimization (TRPO), the model learns to prioritize temporally aligned documents. Furthermore, it employs temporal embedding interpolation to generalize to unseen time periods. The approach demonstrates substantial gains on Temporal QA tasks over strong baselines.

**Compliance With Llm Reviewing Policy:**

Affirmed.

**Key Questions For Authors:**

1. How robust is the embedding interpolation method when faced with sudden, unpredictable real-world events (e.g., a sudden crisis changing the semantic context of a keyword overnight)?
2. Can the authors comment on the computational overhead introduced by the temporal embedding alignments during the large-scale indexing phase?

**Limitations:**

yes

**Strengths And Weaknesses:**

Strengths
- **Addresses a Critical Flaw:** The "temporal hallucination" or mismatch in semantic dense retrieval is a well-known but under-explored issue. The problem formulation is highly motivated.
- **Elegant Adaptation:** Adapting preference optimization (TRPO) to the temporal dimension is intuitively sound and theoretically well-grounded.
- **Strong Generalization:** The use of embedding interpolation to handle unseen timestamps shows good generalization capabilities, making the model practically viable.

Weaknesses
- **Handling Abrupt Concept Drifts:** Temporal interpolation assumes a relatively smooth evolution of semantic meaning. It is unclear how the model performs during sudden events or abrupt concept drifts where interpolation might fail.
- **Scalability of Timestamps:** The continuous representation and interpolation of time might face scalability issues when applied to ext

---

> ### Author Rebuttal · Authors · 2026-03-30
>
> We appreciate your constructive review. We respond to each of your points below.
>
> ---
> > W1/Q1: How robust is the embedding interpolation method when faced with sudden, unpredictable real-world events (e.g., a sudden crisis changing the semantic context of a keyword overnight)?
>
> Thank you for your thoughtful feedback. We respectfully clarify that **handling abrupt events (e.g., a sudden crisis) represents a fundamentally different regime from the general temporal shift setting targeted by our work and the baselines**. Our approach (TPOUR) is designed for general temporal shift, and we show strong generalization in this setting (Tab.1-4, Fig.4).
>
> To fully clarify, we would like to discuss (1) what is different and (2) why separate tailored methods may be needed to handle abrupt drift.
>
> **(1) Abrupt drift differs from general temporal drift [1,4]** because it involves a discontinuous change in the underlying data distribution.
> - By contrast, TPOUR assumes comparatively smoother transitions between nearby periods. Prior surveys [2-4] explicitly distinguish abrupt/sudden drift from general temporal drift.
>
> **(2) As a result, abrupt drift may require separate, drift-aware adaptation methods**.
> - Prior work [2-4] explicitly advocates adapting models based on the type of drift, and recent studies on robust drift handling show that abrupt drifts demand different adaptation or retraining from general temporal drifts.
>
> ---
> >W2/Q2: Can the authors comment on the computational overhead introduced by the temporal embedding alignment during the large-scale indexing phase?
>
> We value your perspective. As you noted, large-scale indexing may introduce computational overhead—especially if multiple interpolated retrievers are involved. However, we clarify that **TPOUR is a framework for temporal alignment in unsupervised retrieval, and a single time-aligned retriever is often sufficient in practice**.
>
> **(1) TPOUR is already practical with a single-model setup.**
> - We show that TPOUR can generalize to future time periods (Tab. 27), allowing users to retrieve the most recent documents without retraining. Similarly, users can leverage past-time-aligned TPOUR retrievers for historical analysis.
>
> **(2) A single TPOUR retriever has the same indexing cost as other retrieval models with the same backbone.**
> - We use a 110M-parameter model and show that it is competitive with much larger embedding models (e.g., 8B, ~72.7× larger) in temporal alignment. Accordingly, **document indexing throughput is expected to be significantly higher (i.e., tens of times faster) for our model**.
>
> Table. Qwen3-Embedding-8B vs. TPOUR Contriever on SituatedQA Explicit.
>
> |  | 2018 N@5 | 2018 N@10 | 2021 N@5 | 2021 N@10 |
> |---|---|---|---|---|
> | Qwen3-Embedding-8B | 29.57 | 33.12 | 34.32 | 37.54 |
> | TPOUR | **44.10** | **46.57** | **40.83** | **41.34** |
>
> **(3) Interpolation is designed to improve training efficiency, not to increase indexing cost.**
> - **The core role of TPOUR interpolation is to enable generalization to intermediate periods by interpolating between retrievers specialized for endpoint periods**. This design is motivated by empirical evidence showing that a single retriever trained to cover all time periods (e.g., TimeR4) does not achieve the optimal performance (Tab.2).
>
> ---
> We hope our response addresses your questions. We also appreciate your acknowledgment of our problem formulation in an under-explored domain, as well as our proposed mitigation approach, TRPO. Please let us know if you have any remaining questions.
>
> [1] A survey on machine learning for recurring concept drifting data streams, Expert Syst. Appl., 2023
>
> [2] Robust concept drift handling in dynamic industrial systems: A multi-objective optimization approach, Knowledge-Based Systems, 2026
>
> [3] Concept Drift Adaptation by Exploiting Drift Type, ACM Trans. Knowl. Discov. Data, 2024
>
> [4] Learning under Concept Drift: A Review, IEEE Transactions on Knowledge & Data Engineering, 2019

---

> > ### Author Rebuttal · Reviewer_qX27 · 2026-04-06
> >
> > Thank you for your detailed answer. After carefully considering your responses, I decided to maintain my original score.

---

> > > ### Author Response · Authors · 2026-04-07
> > >
> > > We sincerely appriciate the reviewer for careful consideration of our responses. We are glad that our rebuttal helped clarify the key aspects of our work. We will incorporate the discussed revisions in the final version to further strengthen the paper.

---

### Decision · Program_Chairs · 2026-04-30

**Decision:**

Accept (regular)

**Comment:**

This work attempts to retrieve semantic meaning that is sensitive to temporal information in an unsupervised fashion. This is achieved by integrating a temporal retrieval method to guide the retriever to favor temporally aligned documents. Experiments show promising results.